# Time varying changes and uncertainties in the CMIP6 ocean carbon sink from global to local scale

Parsa Gooya[1], Neil C. Swart[2,1], Roberta C. Hamme[1]

[1]School of Earth and Ocean Sciences, University of Victoria, Victoria, BC, V8P 5C2, Canada

[2]Canadian Centre for Climate Modelling and Analysis, Environment and Climate Change Canada, Victoria, BC, V8W 2P2, Canada

*Correspondence to*: Parsa Gooya (parsa.g76@gmail.com)

**Abstract.** As a major sink for anthropogenic carbon, the oceans slow the increase of carbon dioxide in the atmosphere and regulate climate change. Future changes in the ocean carbon sink, and its uncertainty at a global and regional scale, are key to understanding the future evolution of the climate. Here we report on the changes and uncertainties in the historical and future ocean carbon sink using output from the Coupled Model Intercomparison Project Phase 6 (CMIP6) multimodel ensemble and compare to an observation based product. We show that future changes of the ocean carbon sink are concentrated in highly active regions - 70 percent of the total sink occurs in less than 40 percent of the global ocean. High pattern correlations between the historical uptake and projected future changes in the carbon sink indicate that future uptake will largely continue to occur in historically important regions. We conduct a detailed breakdown of the sources of uncertainty in the future carbon sink by region. Consistent with CMIP5 models, scenario uncertainty dominates at the global scale, followed by model uncertainty, and then internal variability. We demonstrate how the importance of internal variability increases moving to smaller spatial scales and go on to show how the breakdown between scenario, model, and internal variability changes between different ocean regions, governed by different processes. Using the CanESM5 large ensemble we show that internal variability changes with time based on the scenario, breaking the widely employed assumption of stationarity. As with the mean sink, we show that uncertainty in the future ocean carbon sink is also concentrated in the known regions of historical uptake. Patterns in the signal-to-noise ratio have implications for observational detectability and time of emergence, which we show to vary both in space and with scenario. We show that the largest variations in emergence time across scenarios occur in regions where the ocean sink is less sensitive to forcing - outside of the highly active regions. In agreement with CMIP5 studies, our results suggest that to for a better chance of early detection of changes in the ocean carbon sink, and to efficiently reduce uncertainty in future carbon uptake, highly active regions, including the Northwest Atlantic and the Southern Ocean, should receive additional focus for modelling and observational efforts.

## 1. Introduction

Recent increases in greenhouse gases have trapped additional heat relative to the pre-industrial era and raised Earth's average temperature. Carbon dioxide ($CO_2$) is the primary driver of eglobal warming in the industrial period (Masson-Delmotte et al., 2021). The concentration of atmospheric $CO_2$ has increased from approximately 277 parts per million (ppm) in 1750 (Joos et al., 2008), the beginning of the Industrial Era, to 409 ppm in 2019. However, less than half of the $CO_2$ emitted by anthropogenic activity has remained in the atmosphere. The remaining $CO_2$ was taken up by the natural carbon sinks of the ocean and the terrestrial biosphere. Specifically, the global ocean absorbed ∼26% of the total $CO_2$ emissions during 2011-2020 (Friedlingstein et al., 2021).

The ocean's capacity to absorb anthropogenic $CO_2$ is not uniformly distributed (McKinley et al., 2016, Sarmiento et al., 1998). Despite increasing atmospheric $CO_2$ concentrations, projected air-sea $CO_2$ fluxes do not change much in the middle of the subtropical gyres over the decade starting in 1990. The regions where ocean carbon uptake notably increases are those with strong exchange between the surface and the deep ocean (Ridge and McKinley, 2021; Frölicher et al., 2015; McKinley et al., 2016). The response of the ocean carbon sink to increasing atmospheric $CO_2$ levels consists of a direct absorption response as well as climate change induced perturbations to the natural background carbon fluxes (Crisp et al. 2022, McKinley et al. 2020, Hauk et al., 2020, Gruber et al. 2019, Frolicher at al, 2015). Even within regions there are large variations in the dominant mechanisms and possibly the direction of the carbon sink (or source). In the Southern Ocean, for instance, the spatial superposition of natural and anthropogenic $CO_2$ fluxes leads to a relatively strong uptake band between approximately 55°S and 35°S (Gruber et al., 2019). However, south of the Polar Front (55°S), the different estimates agree less well (Gruber et al., 2019, Landschützer et al., 2016, Gruber et al., 2009, Takahashi et al., 2009). Supported by measurements on biogeochemical floats (Bushinsky et al., 2019; Gray et al., 2018; Williams et al., 2018), Gruber et al. (2019) argue that the region was most likely a small source at the time.

Earth System Models (ESMs) are the primary tool for projecting the future evolution of carbon in the climate system. However, quantitative projections from ESMs are subject to considerable uncertainty, particularly at regional and local scales (Friedrich et al., 2012; Frölicher et al., 2014; Hauck et al., 2015; Roy et al., 2011; Tjiputra et al., 2014; Terhaar et al., 2021) where less averaging is done and different individual mechanisms dominate different regions. Projection uncertainty varies with lead time, spatial averaging scale, and from region to region (Lovenduski et al., 2016; Schlunegger et al., 2020). For example, Lovenduski et al. (2016) showed a spatially

heterogeneous pattern of projection uncertainty in $CO_2$ flux projections over 17 ocean regions for CMIP5 models. Furthermore, by comparing uncertainty at the global scale to the scale of the California Current System, they show that uncertainty is higher at smaller scales. Schlunegger et al. (2020) further show partitioning of uncertainty for 10 ocean basins at the year 2050. All said, if ESMs are to be used to quantify future changes in ocean carbon uptake, especially across shorter timescales and at regional spatial scales, and to inform observational campaign planning, their uncertainties must be well known and well understood (Lovenduski et al., 2016).

A systematic characterization of projection uncertainty has become possible with the advent of the Coupled Model Intercomparison Project (CMIP), as a number of climate models of similar complexity provided simulations over a consistent time period and with the same set of emissions scenarios (Lehner et al., 2020). There are three main types of uncertainty in climate model projections, as described by Hawkins and Sutton (2009) (hereafter HS09):

**Uncertainty due to internal variability:** Internal variability is the unforced natural climate variability resulting from the internal processes in the climate system. Modes such as the El Niño–Southern Oscillation, North Atlantic Oscillation, Atlantic Multidecadal Oscillation, Pacific Decadal Oscillation, and Southern Annular Mode (SAM) contribute to this internal variability. Internal variability also includes variability that acts on shorter time and spatial scales, such as submesoscale and mesoscale ocean features (Frolicher et al., 2016). The real world follows only one of an infinite possible number of *realizations* of internal variability, and due to its chaotic nature, the future evolution of internal variability is not predictable beyond short timescales (Lorenz, 1969; Somerville, 1987). Climate model simulations do not attempt to reproduce the exact observed evolution of internal variability, but produce their own, unique realizations that aim to capture the statistics of variability. Hence, our analysis must account for internal variability, both when comparing historical model simulations to observations, and when considering uncertainties in the future ocean carbon sink. In HS09, a fourth-order polynomial fit to simulated global and regional temperature timeseries represented the forced response, while the residual from this fit represented the internal variability. There is thus, an assumption of stationarity (constant in time) in their method. Moreover, this approach could possibly conflate internal variability with the forced response in cases where low-frequency (decadal-to-multidecadal) internal variability exists, or when the forced signal is weak, which makes the statistical fit a poor estimate of the forced response (Kumar and Ganguly, 2018). In this study, we instead use a Single-Model Initial-condition Large Ensemble (SMILE) to robustly quantify the internal variability across time and scenarios using ensemble statistics (Lehner et al., 2020). A SMILE is an ensemble of model realizations that each starts from

different initial conditions but uses the same model and forcing, and provides representations of the climate system that are equivalent except for internal variability.

**Uncertainty due to model structure:** Models differ in their resolution, structure, numerics, and parameterization of processes. These differences cause models to respond differently to the same forcing. For example, the CMIP5 model simulations run under Representative Concentration Pathway 8.5 (RCP8.5) project a wide range of cumulative anthropogenic carbon storage by 2100 (320–635 Pg-C) (Ciais and Sabine, 2013) due to both internal variability and model uncertainty (Lovenduski et al., 2016).

**Uncertainty due to emission scenario:** The future of the climate system depends on human activity and our emission of climate active gases that change radiative forcing. Future emissions are highly uncertain, given our inability to project the complex changes in society and technology upon which they depend. As a result, future simulations are run with a range of possible "scenarios" for how future emissions (or atmospheric concentrations) will evolve under different socioeconomic storylines. These scenarios are prescribed via the internationally coordinated experiments organized by the Coupled Model Intercomparison Project (O'Neill et al., 2016). Since the future emission trajectory is unknown, these future simulations are referred to as projections, rather than predictions. Projections of future ocean carbon uptake from ESMs are greatly influenced by the choice of emission scenario (Lovenduski et al., 2016). For example, cumulative ocean carbon uptake from 1850 is projected to saturate at approximately $290 \pm 30$ GtC under ssp126, and to reach $520 \pm 40$ GtC by 2100 under ssp585 for CMIP6 models (Canadell et al., 2021).

Together with the patterns of changes in the sink, the patterns of internal variability allow for an assessment of the required timescales for detection of changes in the ocean carbon sink. Detection means that we can robustly separate the forced signal from internal variability (McKinley et al., 2016). Detectability can be assessed using Time of Emergence (TOE; Hawkins and Sutton, 2012; Lovenduski et al., 2016; McKinley et al., 2016; Rodgers et al., 2015; Schlunegger et al., 2020; 2019). For example, McKinley et al. (2016) and Schlunegger et al. (2019) showed that the forced signal of increasing ocean carbon uptake is not detectable in regions of convergent Ekman transport (centre of the subtropical gyres). Schlunegger et al. (2020) builds on that using four large ensembles of CMIP5 ESM simulations with two forcing scenarios to show that air-sea $CO_2$ flux TOEs show strong agreement between the large-ensembles not just for global and regional scales but also locally and spatially. Their use of only four models and two scenarios however, potentially underestimates the contribution of model and scenario uncertainty.

Here, we build on previous work using CMIP6 models. We make use of an ensemble of 13 models to better capture
model uncertainty in the response to different forcing (scenarios) and three scenarios to represent a wider range of
future possibilities including a strong mitigation scenario. We start by analysing the regional patterns of historical
ocean carbon uptake and how they are projected to change in the future (Sect. 3.1). We estimate internal variability
from a comprehensive SMILE, avoiding the stationarity assumption common in previous work, which we show is
violated. Then, we examine the partitioning among different sources of uncertainty (Sect. 3.2) and provide a novel
analysis of how the three sources of variability change across the full continuum of scales (Sect. 3.3). Having
shown how the uncertainty and distribution among sources differ based on scale of integration and region of
interest, we analyze local patterns of uncertainty by source (Sect. 3.4). The final section explores the detectability
of the model projected signal given the uncertainty imposed by internal variability. We report on the scenario-
dependent Time of Emergence, using a scenario specific measure of internal variability in order to make useful
suggestions for future observations.

**2. Data and Methods**
2.1 Model Data Selection
Here we use results from models selected from the 6[th] Coupled Model Intercomparison Project (CMIP6; Eyring
et al., 2016). Models are chosen based on availability, meaning all models that provided at least one realisation
for air-sea $CO_2$ flux (fgco2) for the $CO_2$ concentration driven experiments of interest. One realization of each
model over the historical period and three scenarios that represent the low (ssp126), mid (ssp245), and high
(ssp585) ranges of future atmospheric $CO_2$ concentrations are analysed. A total of 16 models met these criteria,
out of which 3 were excluded as outliers (see section S1 in the Supplements). To maintain equal sampling, only
one realization of each model was selected, except when specifically using the large ensembles to assess internal
variability. Finally, since the ocean component of the models may be on different grids, all model data were
remapped to a regular one-by-one-degree grid and a 10 year running mean filter was applied to the time-series.
We did not account for potential drift in the models. However, the drift is known to be small in the models
compared to the historical trends for CMIP5 models (Hauck et al, 2020). For 11 of our CMIP6 models for which
piControl runs are available, on average, the drift is more than one order of magnitude smaller than the change in
the model scenario with the smallest trend over the 21st century, on the global scale.

## 2.2 Sources of uncertainty

Total uncertainty is composed of internal, model, and scenario uncertainty in equation 1, which assumes that each
of these sources is independent. Here, each source of uncertainty is considered as a function of time ($t$) and location
($l$) (Lovenduski et al., 2016):

$$U_T{}^2(t, l) = U_I{}^2(t, l) + U_M{}^2(t, l) + U_s{}^2(t, l) \tag{1}$$


where $U_T(t, l)$ is total uncertainty, $U_I(t, l)$ is internal variability, $U_M(t, l)$ is model uncertainty, and $U_S(t, l)$ is
scenario uncertainty. The fractional uncertainties for each source are calculated as $\frac{U_I^2}{U_T^2}$, $\frac{U_M^2}{U_T^2}$, and $\frac{U_S^2}{U_T^2}$ (Lovenduski et
al., 2016).

HS09 assume $U_I(t, l)$ to be constant in time (stationary) and use a 4$^{th}$ degree polynomial fit to measure internal
variability as the spread over time and scenario of the residuals for each model's signal relative to the fitted signal.
We show in the Supplements (see section S2) that internal variability depends on time and scenario, violating the
commonly used assumption of stationarity. Using a SMILE allows us to account for these variations without having
to make assumptions about distribution or stationarity of variability (Frolicher et al., 2015; Schlunegger et al.,
2020). Here we estimate internal variability as two times the standard deviation of the annual carbon sink across
50 realizations from a SMILE based on CanESM5 (Eq. 2):


$$U_I(t, l) = 2\sqrt{\frac{1}{Ns}\sum_{s=1}^{N_s} \text{Var (CanESM5 Large Ensemble)}} \tag{2}$$


where $s$ indicates each scenario ($Ns$ is the number of scenarios) and Var indicates the variance over the large
ensemble of CanESM5. In the CanESM5 SMILE, each realization starts from different initial conditions which
are drawn from points separated by 50 years in the piControl simulation. Thus, the spread across the realizations
gives a robust estimate of the internal variability, including sampling over longer term ocean variability.

Previous studies have also used SMILEs to estimate variability (Frolicher et al., 2015; Schlunegger et al., 2020),
although they used either a limited ensemble size or single scenario. We show in the Supplements (Fig. S2), that a
sufficiently large ensemble size is needed to capture internal variability, and that internal variability depends on the
scenario. In the ideal case, if every CMIP model provided sufficiently large SMILEs for each scenario, an ensemble
mean estimate of the variability could be obtained and would represent a best estimate (but still possibly biased
compared to the real world). However, only a handful of CMIP6 models produced multiple ensemble members.
We selected the CanESM5 SMILE as it is the only model that has a large enough ensemble over the entire timeline
and set of experiments to estimate internal variability robustly across scenarios.

The use of a single model to estimate the scale of internal variability leads to some uncertainty in our estimates, as
models do not agree perfectly with each other on the variability. Nonetheless, over the historical period, variability
among large ensembles from three models that have enough ensemble members is within 10%, on the global scale
(Fig S3). Differences will be larger at smaller scales; however, the general patterns of the magnitude of internal
variability (see Fig. S4) are in good agreement across models and are consistent with known regions of high
variability in the observed ocean, validating our use of the CanESM5 SMILE

Model uncertainty is calculated by taking the variance across the forced signal of all available models for each
scenario, averaging over the three scenarios, and then reporting twice the square root of the result (Eq. 3).

$$U_M(t,l) = 2\sqrt{\frac{1}{Ns}\sum_{s=1}^{N_s} \text{Var}_m\big(F(m,s,t,l)\big)} \qquad (3)$$

where $\text{Var}_m$ means the variance taken across different models ($m$) for individual times and scenarios ($s$).
$F(m,s,t,l)$ is the forced signal and can be related to each realization as follows:

$$T(m,s,t,l) = F(m,s,t,l) + R(m,s,t,l) \qquad (4)$$


Where, $T(m,s,t,l)$ represents the reported output, i.e. each realization, but must be corrected for internal
variability. $R(m,s,t,l)$ is the residual from the forced signal caused by internal variability. Here, the variance in
the forced signal across all models is calculated by correcting the total variance across all models' one realization
for the variance caused by internal variability. The corrections are done by subtracting the variance across the same
number of CanESM5 ensemble members as the multi-model ensemble (13 members) from the variance across the
one realization of each of the 13 models. For this correction only, the sample sizes (13) are kept the same so that
the internal variability removed from the variance across the models' first realizations is not overestimated by a
well sampled 50-member ensemble (see section S3 in the Supplements).

$U_s$ (t, l) is the scenario uncertainty. Scenario uncertainty is measured as twice the standard deviation (square root
of variance) across scenarios of the multi-model mean signal (Eq. 5).

$$U_S(t,l) = 2 \sqrt{\text{Var}_m \left( \frac{1}{N_m} \sum_{m=1}^{N_m} T(m,s,t,l) \right)} \qquad (5)$$

where $N_m$ is the number of models. The multi-model mean across the first realizations of the 13 models is an
estimate of the multi-model forced response and does not require correction for internal variability as done for
model uncertainty.

We conduct analysis on three different scales: single grid point (one-degree resolution), regional, and global. When
regional and global analysis is done, the dependence on location is taken away by averaging over that region or the
whole global ocean.

2.3 Time of Emergence (TOE)
In order to know when the forced response is distinguishable from internal variability, TOE is calculated. The time
of emergence is the first year when the multi-model mean anomaly is larger than internal variability – approximated
by two times the standard deviation across the 50 member CanESM5 ensemble - for five consecutive years (the
first year of this five-year period is reported as the time of emergence). The result is reported at each grid point for
the 10-year running mean smoothed anomaly relative to the 1995-2015 mean (detection of a change relative to the
current state of the ocean).

2.4 Scale Dependence
Finally, the scale dependence of the sources of uncertainty is measured at year 2050 using ssp245 for internal
variability and model uncertainty, and using all scenarios for scenario uncertainty. The analysis is done by moving
a sliding sample window of a given area across the earth, and then repeating with a larger and larger window until
all scales from <100 km$^2$ to the whole Earth are considered. For each source of uncertainty and averaging scale,
the average for all rectangles across the globe is reported, where each rectangle contains the same ocean area.

**3.  Results and Discussion**
3.1 Global Analysis
The pattern of the carbon sink in the CMIP6 multi-model ensemble mean from the historical experiment over 1995-
2015 matches that of the Landschützer (2016) Self Organizing Map - Feed Forward Neural Network (SOM-FFN)
observation-based data product estimate (correlation coefficient of 0.84, compare Figs. 1a and 1b). We use the
multi-model mean response to external forcing as a more robust estimate of the forced climate signal than the
response of any single model (Tebaldi & Knutti, 2007). Unlike in ESMs, the observation-based product only
represents the one realization of the real world, which includes internal variability, and is therefore not directly
equivalent to the forced signal. However, the comparison to the 20 year mean multi-model mean still informs us
about the degree of agreement between the two products. When compared to the observation-based data product,
the CMIP6 multi-model mean shows a larger sink (positive flux) in the North Atlantic and North/North-West
Pacific but a smaller sink in the Southern Ocean (Fig 1a, b). Additionally, the observation-based data product shows
a larger source in the Equatorial Pacific and Indian Ocean than the CMIP6 multi-model ensemble.

While most of the global ocean shows a net sink relative to the pre-industrial era, the largest acceleration of that
sink takes place in some highly active regions such as the subpolar North Atlantic, Southern Ocean, Eastern
Equatorial Pacific, and western boundary currents of the mid-latitude gyre systems in the Pacific and Atlantic
Oceans (Fig. 1c). These regions of largest change in the carbon sink (direct response to higher atmospheric $CO_2$
plus changes in the natural carbon sink) are the regions where there is a surface-depth connectivity through ocean
circulation as the air–sea flux of anthropogenic carbon is fundamentally limited by the rate of surface-to-depth
transport (Graven et al., 2012; Ridge and McKinley 2021). These results for CMIP6 models are consistent with
those from McKinley et al. (2016) based on CESM-LE under CMIP5 protocols, and earlier studies such as
Sarmiento et al. (1998). Here, we provide a new criterion for identifying these highly active regions based on
comparing the integrated global sink anomaly within grid cells above a certain threshold to the percentage of ocean
area they occupy (see Supplement S5). We find that for all three scenarios and both mid-21$^{st}$ century (2040-2060
mean) and late-21$^{st}$ century (2080-2100 mean) time periods (with the exception of ssp126 late-century where strong
mitigation of anthropogenic $CO_2$ emissions results in broad patterns of negative anomalies), approximately 70% of
the changes in the sink relative to the preindustrial era take place in less than 40% of the global ocean (see
Supplement Fig. S6 and S7). The diagnosed highly active regions based on this analysis (Fig. S7) are consistent
with the regions of large uptake change (trends) from previous studies (Rodgers et al., 2020; McKinley et al., 2016;
Frölicher et al., 2015)

The regions of largest future carbon uptake, relative to the 1995-2015 mean, are within the same highly active
regions responsible for most of the uptake over the historical period. The correlation coefficients at the top of each
panel in Fig. 1 (except 1b) represent the pattern correlation between future absolute anomalies, relative to 1995-
2015, and anomalies in 1995-2015, relative to the pre-industrial era. The high correlations indicate that regions that
have been most active in increasing their carbon sequestration are the same regions that will continue to increase
further into the future, particularly with larger increases in atmospheric $CO_2$ (ssp585).  Our results support the
findings of Wang et al. (2016) who showed that projected future air-sea $CO_2$ fluxes are strongly associated with
simulated historical air-sea $CO_2$ fluxes. This confirms that the historical state is a good predictor for the future state
(Wang et al., 2016) not only in terms of magnitudes of the sink, but also in the spatial pattern.



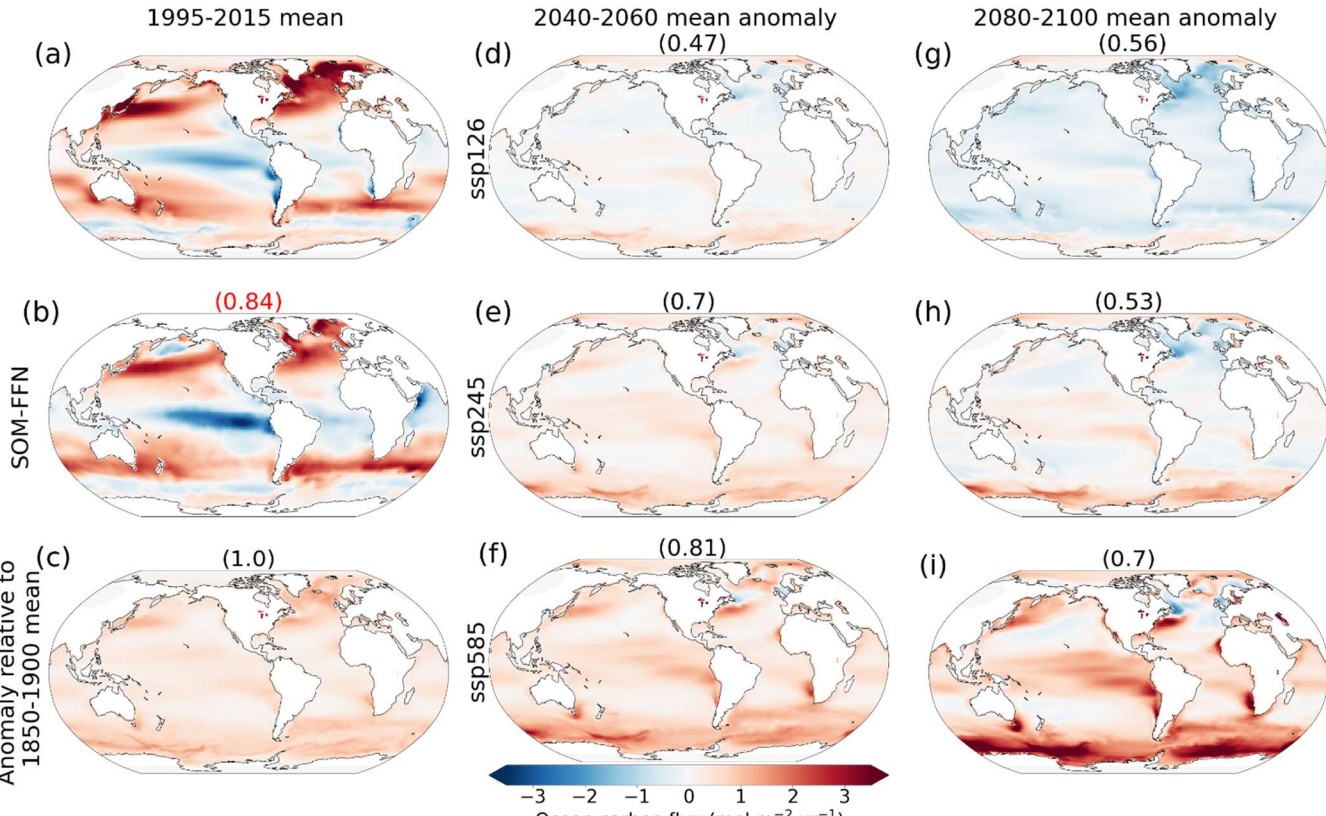


**Figure 1**- CMIP6 multi-model mean maps of carbon sink and sink anomalies using one realization of each model. Columns
represent different time periods, being the recent time (1995-2015 mean), mid-century (2040-2060 mean), and late-century
(2080-2100 mean). Note: the sink is positive into the ocean.  The first column shows (a) the CMIP6 ensemble mean air-sea
$CO_2$ flux over 1995-2015, (b) Landschützer et al. (2016) SOM- FFN product, and (c) the CMIP6 ensemble mean flux
anomaly over 1995-2015 relative to the 1850-1900 mean. Other panels are anomalies relative to the 1995-2015 multi-model
mean (panel a). Panels d through i show different scenarios. Numbers above each map are correlation coefficients between
the absolute value of the change relative to 1995-2015 with the 1995-2015 anomaly map relative to the pre-industrial era in
panel c, except the red number at the top of panel b that is the correlation coefficient with this panel and panel a.

The multi-model mean sink anomalies for two future periods, 2040-2060 and 2080-2100, show how the sink is
projected to evolve, relative to 1995-2015, according to time and choice of emission scenario (Fig. 1d-i). The
regional patterns show mostly positive anomalies at mid-century with largest changes in the higher emission
scenarios (ssp585). Towards the end of the century, however, greater areas of negative anomalies are expected in
ssp126, as emissions turn negative in the late 21st century in this scenario. The largest absolute values of anomalies
are still within the same highly active regions discussed before with surface-depth connectivity regardless of it

being positive or negative. The late-century anomalies are predominantly positive in ssp585 which corresponds to the highest emission scenario (continuing to grow larger compared to the mid-century), while ssp245 is somewhere in between, with regions of positive and negative anomalies. Under ssp245, as $CO_2$ emissions decrease and atmospheric $CO_2$ start to level off, the intensity of uptake decreases in the midlatitude western boundary currents and subpolar North Atlantic in the late-century, and anomalies in the Eastern Equatorial Pacific also decrease, compared to the mid-century. The globally integrated ocean carbon uptake anomaly rates are summarized in Table 1.

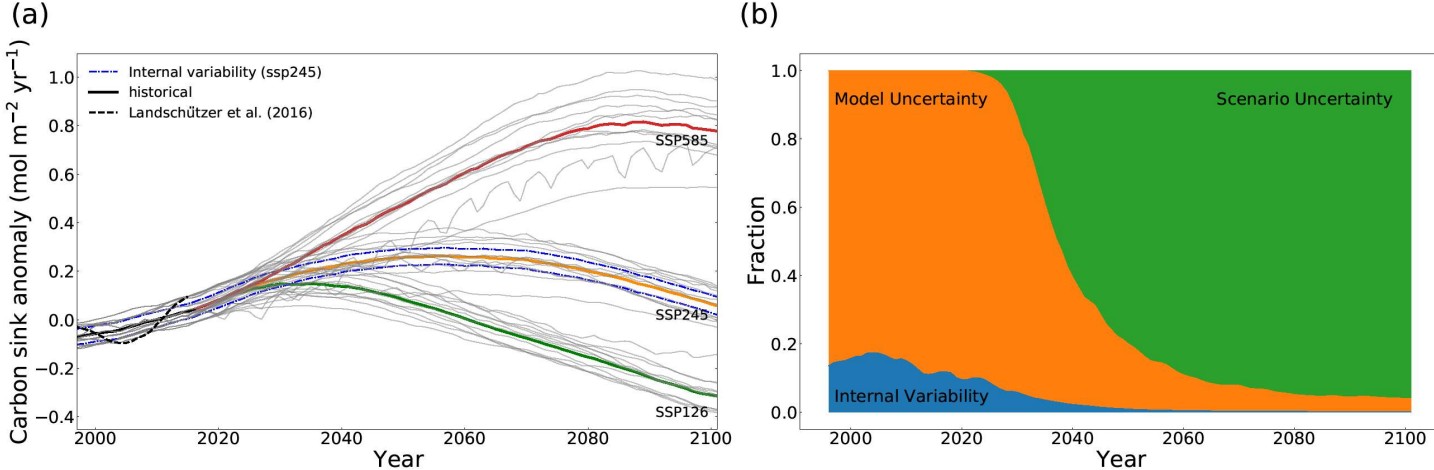

**Figure 2**- (a) Thick lines are multi-model means of the global mean ocean carbon sink anomaly timeseries relative to 1995-2015. Individual models are plotted as thin grey lines in the background. The black dashed line shows the Landschützer et al. (2016) SOM-FFN product. Both models and SOM-FFN timeseries are smoothed with a 10-year running mean. The blue dashed lines show internal variability for ssp245. (b) Timeseries showing the breakdown of uncertainty to different sources with time for the global ocean carbon sink anomaly. The internal and model uncertainty are averaged for different scenarios.

| | Scenario | 1995-2020 | 2020-2040 | 2040-2060 | 2060-2080 | 2080-2100 |
|---|---|---|---|---|---|---|
| **Anomaly (range)** | **ssp126** | 0.00 (-0.06 – 0.06) | 0.13 (0.05 – 0.21) | 0.07 (-0.02 – 0.16) | -0.08 ( -0.14 - -0.01) | -0.24 (-0.3 - -0.12) |
| | **ssp245** | | 0.17 (0.08 – 0.24) | 0.25 (0.11 – 0.36) | 0.23 (0.09 – 0.33) | 0.13 (0.02 – 0.21) |
| | **ssp585** | | 0.22 (0.11 - 0.30) | 0.49 (0.29 – 0.62) | 0.71 (0.45 – 0.90) | 0.80 (0.54 – 1.00) |
| **Internal (model) Uncertainty** | **ssp126** | 0.032 (0.08) | 0.033 (0.11) | 0.034 (0.11) | 0.035 (0.10) | 0.036 (0.11) |
| | **ssp245** | | 0.032 (0.11) | 0.034 (0.14) | 0.037 (0.14) | 0.036 (0.12) |
| | **ssp585** | | 0.033 (0.13) | 0.037 (0.2) | 0.045 (0.26) | 0.043 (0.27) |
| | **Average** | 0.032 (0.08) | 0.033 (0.12) | 0.035 (0.16) | 0.039 (0.18) | 0.038 (0.18) |


**Table 1**- CMIP6 multi-model mean globally averaged carbon sink anomalies (with ranges within the 20-yr period in
parentheses) relative to the 1995-2015 mean (in mol-C m$^{-2}$ yr$^{-1}$) and internal variability from CanESM5 (with model
uncertainty in parentheses) for the globally averaged ocean carbon sink anomalies for the three scenarios and the average
values across scenarios.

The trends in the global mean ocean carbon sink anomalies over 1995-2015 are statistically consistent between the
CMIP6 multi-model ensemble mean and the Landschützer et al. (2016) observation-based data product (Fig. 2-a),
based on the test from Santer et al. (2008; see Supplements section S5). However, the SOM-FFN based time-series
shows a larger multi-decadal variability (variations in the 10-year running mean timeseries on top of the trend) than
seen in individual model realizations, and is larger than the range of internal variability estimated from the
CanESM5 SMILE. The difference could be due to either overestimation of internal variability by the SOM-FFN
method, or underestimation of the internal variability by the ESMs. Given that on regional scales the SOM-FFN

data is within the range of internal variability projected by the CMIP6 large-ensemble of CanESM5 (see Sect. 3.3), and that there are significant gaps in the spatial and temporal sampling that underlies the Landschützer et al. (2016) estimate, it seems plausible that the discrepancy is largely due to overestimation of internal variability on the global scale by the SOM-FFN technique. This is consistent with the findings of Gloege et al. (2021), which showed that, globally, the magnitude of decadal variability is overestimated by 21% by the SOM-FFN technique, attributed to the amount of data filling.

On the global scale, model uncertainty is the dominant source of uncertainty in the historical period, but scenario uncertainty comes to dominate later (Fig. 2b). Over the 1995-2020 period, model uncertainty explains around 85% of the total uncertainty. Scenario uncertainty becomes the dominant source after 2040, explaining almost 40% of the total uncertainty at that time and more than 90% by the end of the century. Internal variability explains 15% at the start of the century but only around 1% by the end. It is worth mentioning that the decreased share of uncertainty associated with model and internal variability do not mean that model or internal variability decrease in an absolute sense; rather, their importance relative to scenario uncertainty declines. These results regarding the importance of model and scenario uncertainties for multidecadal projections, and dominance of scenario uncertainty with time agree with previous studies using CMIP5 models (Lovenduski et al., 2016; Schlunegger et al., 2020).

Absolute internal and model uncertainty of the global carbon sink change with time, based on the scenario (Table 2, Fig. S3). High emission scenarios such as ssp585 show a larger change for both internal and model uncertainty where the forcing is stronger (Fig. S3). When averaged for the three scenarios, a constant increase in the magnitudes of both model and internal variability is seen through the century until 2080-2100 when the values either do not change or decrease slightly (Table 1). Model uncertainty more than doubles towards the end of the century compared to 1995-2015 on average for different scenarios. This is consistent with Lovenduski et al. (2016) who argue that the increase is due to differences in climate sensitivity among models that manifest more strongly with time (and hence cumulative emissions). Additionally, the dependence of internal variability on the scenario is an interesting result. Future SMILEs from multiple models will allow evaluation of the degree of dependence and the driving mechanisms of such changes with time based on the forcing (scenario). Our result of internal variability dependence on scenario implies that the time of emergence of a signal out of internal variability will be affected by changes in the internal variability under different future forcing scenarios – which we return to in Section 3.4.

3.2 Dependence of the sources of uncertainty on spatial scale
It is generally accepted that uncertainty and, most importantly, internal variability grow larger as the averaging
(integration) scale gets finer, because on larger scales the variability is averaged out. Here, we provide a novel and
continuous view of change in variability across scales from the global to grid scale, by measuring how variability
changes relative to scale on average (Fig. 3). At the global scale, the dominant source of uncertainty is scenario
uncertainty, followed by model and internal variability respectively, consistent with Fig. 2b. However, as the
averaging (integration) scale gets finer, model and internal variability grow rapidly, while scenario uncertainty only
grows slightly on average (over all regions of this size). At an averaging (integration) scale with an area finer than
75 million $km^2$ (on average), model uncertainty becomes the dominant source of uncertainty, and at a scale finer
than 3 million $km^2$, internal variability becomes larger than scenario uncertainty. The idea of scale dependence of
these uncertainties was tested in Lovenduski et al. (2016) by comparing an area covering the California Current
System with the global ocean. Here, we provide a novel analysis on a continuum of scales covering global to
regional to local scales. While the results here hold true on average over the global ocean, scale dependence is
partially controlled by the particular region being sampled. Finally, while our estimates of the magnitudes of
sources of uncertainty and the cross over points (at which the dominance of internal variability over model
uncertainty and model uncertainty over scenario uncertainty takes place), depend on the choice of ESMs and the
method for calculation of internal variability, the general patterns are unlikely to be model dependent.


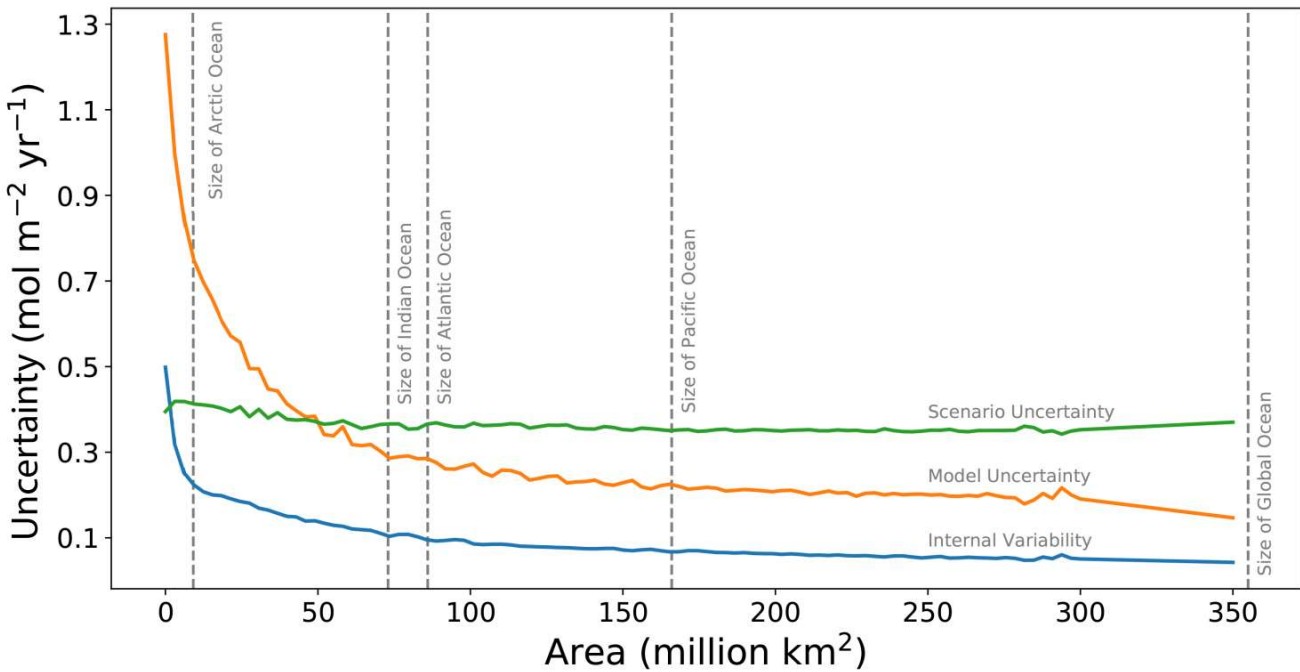


**Figure 3**- Sources of uncertainty versus area of averaging. Internal variability is based on ssp245 year 2050 of all CanESM5
members. Scenario uncertainty is based on all scenarios of the 13 models at year 2050 and model uncertainty is the corrected
standard deviation of our 13 models at year 2050 of ssp245. The values of uncertainties are averaged over all different
rectangular areas of each size that can scan the globe. Dashed lines indicate the size of the averaging window and not a
specific location.


3.3 Regional Analysis
We further expand on the findings of our analysis of the scale dependence of uncertainty averaged over the globe
by repeating the uncertainty breakdown for two specific regions: one in the Northeast Pacific (NE Pacific) between
130°- 160° W and 40°- 60° N, and one in the Northwest Atlantic (NW Atlantic) between 40°- 70° W at the same
latitude.  We chose these regions, first, to be of similar size, and second to represent very different carbon processes.
The NW Atlantic region represents a highly active region while the NE Pacific region is more typical of quiescent
ocean regions, where the flux anomalies are relatively small.

The variation across scenarios is at all times smaller than internal variability in the NE Pacific (Fig. 4a). This
suggests both that it will be difficult to robustly detect any human induced changes in observations of the NE
Pacific carbon sink, and that potential future differences relating to choice of mitigation scenarios will not be
readily apparent in the NE Pacific carbon flux. This is true even for the high emission scenarios, because the
anomalies are small regardless of scenario (Table 2). We speculate that in the absence of mechanisms providing a
pathway to the depth where significant $CO_2$ accumulation occurs, the surface $pCO_2$ trend will follow that of the
atmosphere closely, causing $\Delta pCO_2$ and therefore air-sea carbon flux to remain fairly constant for all scenarios. In
the NW Atlantic however, the variation across scenarios becomes larger than the internal variability in the early
2060s (Fig. 4c). The response of the region to climate change is dependent on the scenario (Table 2), or, in other
words, the amount of carbon dioxide in the atmosphere. This is because the NW Atlantic is a highly active region
where the air-sea flux actively responds to the atmospheric $CO_2$ concentration. The connection to depth allows
for surface water to be replaced with water masses whose $pCO_2$ trend lags behind that of atmosphere. The trend
of the CMIP6 multi-model time-series over the historical period is statistically consistent (See Supplements
section S5) with that of the observation-based SOM-FFN product, and the multi-decadal variability is within the
range of internal variability measured by the CanESM5 large-ensemble in both regions. We note that both of
these regions are relatively well sampled, which may lead to more robust estimates of multi-decadal variability in
the Landschützer et al. (2016) dataset, and better agreement with the models than seen at the global scale.

Fractional estimates of each source of uncertainty vary with time and have different patterns for these two regions.
Internal variability and model uncertainty in the NE Pacific and NW Atlantic are larger by an order of magnitude
than at the global scale (Table 2). A lesser importance for scenario uncertainty and greater importance for internal
and model uncertainty is apparent in both regions compared to the global scale, in agreement with Schlunegger et
al. (2020). Over the period 1995-2020, model uncertainty is the dominant source of uncertainty in both the NE
Pacific and NW Atlantic (80-90%), while the remainder is internal variability (Fig. 4bd). Internal variability
explains around 25-30% of the total uncertainty in the NE Pacific throughout the century. In the NW Atlantic
however, its share drops to 15% by the end of the century. The share attributable to internal variability is much
larger during the 21$^{st}$ century in both regions compared to the global scale. Internal variability is larger in the NW
Atlantic in an absolute sense (Table 2), but its share of the total uncertainty is larger in the NE Pacific (Fig. 4b).
The large share of internal variability in NE Pacific indicates the need for sustained observations in the region.
Overall, internal variability averaged over the scenarios shows a small increase, but no clear trend in time in both
regions until the 2080-2100 period where it decreases, consistent with the global estimates (Table 2). We showed
earlier that in the NE Pacific scenarios do not differ because the region is not a highly active region (Fig. S7) -
scenario uncertainty explains less than 20% of the total uncertainty at the end of the century in the NE Pacific. In
the NW Atlantic, scenario uncertainty grows larger with time, becoming 45-50% of total uncertainty by the end of
the century.  In both regions, model uncertainty is the dominant source of uncertainty in all years.

Our regional analysis confirms that while uncertainty and its distribution among sources depends on the spatial
scale of integration, the specific location also matters (Lovenduski et al, 2016; Schlunegger et al., 2020).
Schlunegger et al., (2020) tested this idea for 10 ocean basins of variable size (see their Figure 9). We focused on
keeping the sizes similar and analyse a highly active region versus a more quiescent ocean region. The key message
here that there is an association with the importance as well as the magnitude of sources of uncertainty with how
active the region is in regards to the carbon sink is not sensitive to the use of CanESM5 for estimation of internal
variability. Local patterns of uncertainty broken down by source are thus needed to clarify changes based on
location.







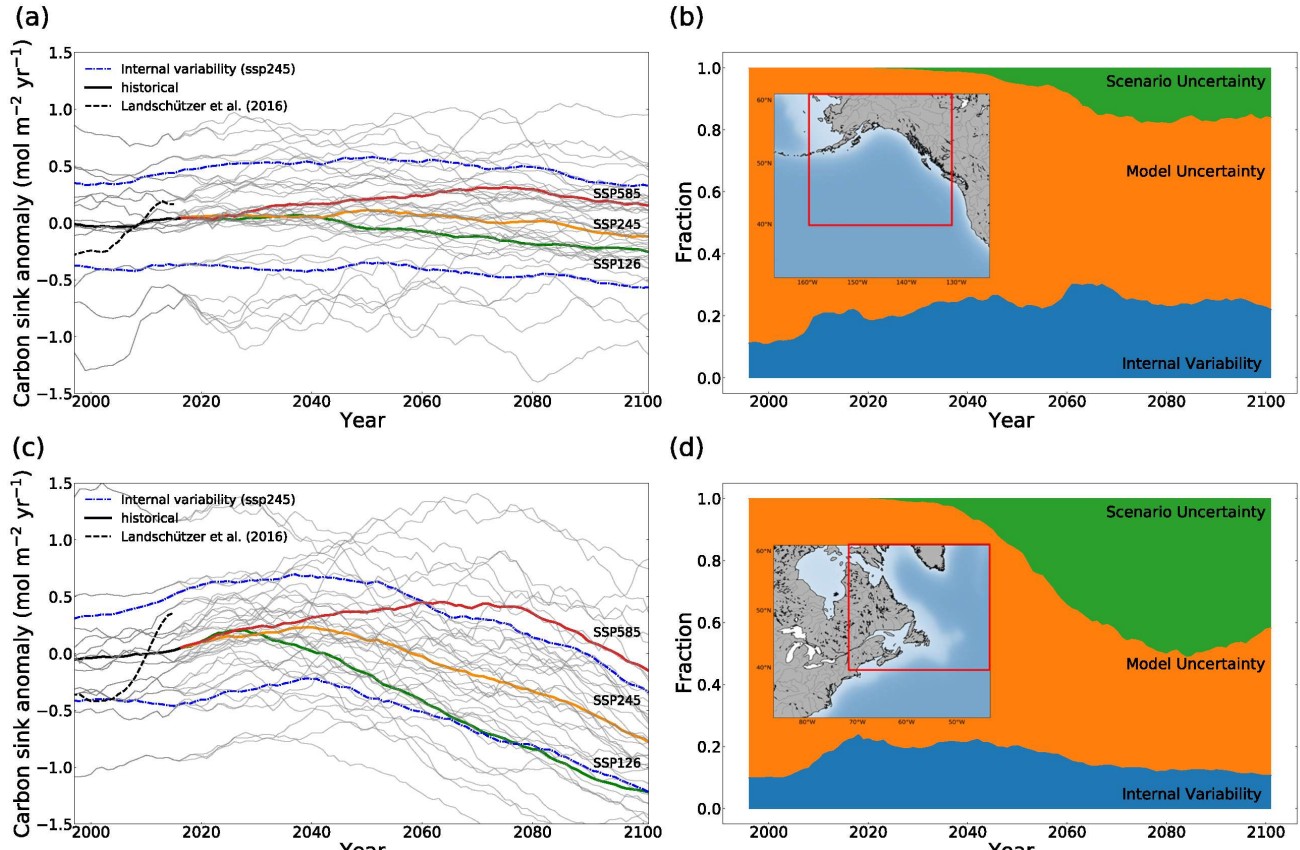


**Figure 4**- (a), (c) Thick lines are multi-model mean timeseries of anomalies relative to the 1995-2015 mean. All model time-
series averaged for the means are plotted in grey lines in the background. The black dashed line shows the Landschützer et
al. (2016) SOM-FFN product.  The blue dashed line shows the internal variability measured as two times the standard
deviation across all 50 members of the CanESM5 SMILE only for ssp245 here. (b), (d) time-series showing the breakdown
of uncertainty to different sources with time. The internal and model uncertainty are averaged for different scenarios. (a), (b)
NE Pacific (40-60 °N, 130 -160 °W). (c), (d) NW Atlantic (40 - 60 °N, 40 -70 °W)







| | | Scenario | 1995-2020 | 2020-2040 | 2040-2060 | 2060-2080 | 2080-2100 |
|---|---|---|---|---|---|---|---|
| **NE Pacific** | **Anomaly (range)** | **ssp126** | | 0.05 (-0.91 – 0.86) | 0.03 (-0.86 – 0.62) | -0.13 ( -1.1 – 0.58) | -0.21 (-1.18 - 0.60) |
| | | **ssp245** | 0.00 (-0.98 – 0.76) | 0.06 (-0.86 – 0.83) | 0.09 (-0.74 – 0.81) | 0.03 (-0.65 – 0.60) | 0.06 (-0.70 – 0.53) |
| | | **ssp585** | | 0.11 (-0.73 - 0.79) | 0.21 (-0.61 – 0.86) | 0.29 (0.22 – 0.94) | 0.2 (-0.25 – 0.98) |
| | **Internal (model) Uncertainty** | **ssp126** | | 0.47 (0.87) | 0.43 (0.74) | 0.40 (0.81) | 0.39 (0.83) |
| | | **ssp245** | 0.39 (0.90) | 0.46 (0.87) | 0.47 (0.81) | 0.48 (0.64) | 0.45 (0.53) |
| | | **ssp585** | | 0.45 (0.81) | 0.47 (0.745) | 0.58 (0.55) | 0.44 (0.57) |
| | | **Average** | 0.39 (0.90) | 0.46 (0.86) | 0.46 (0.77) | 0.47 (0.70) | 0.43(0.67) |
| **NW Atlantic** | **Anomaly (range)** | **ssp126** | | 0.13 (-0.77 – 1.21) | -0.20 (-1.03 – 0.56) | -0.66 ( -1.45 – -0.11) | -1.00 (-1.80 - -0.56) |
| | | **ssp245** | 0.00 (-0.97 – 1.31) | 0.18 (-0.78 – 1.23) | 0.10 (-0.68 – 0.80) | -0.20 (-0.97 – 0.50) | -0.54 (-1.22 – 0.07) |
| | | **ssp585** | | 0.23 (-0.70 – 1.20) | 0.38 (-0.41 – 1.12) | 0.41 (-0.27 – 1.29) | 0.10 (-0.70 – 0.96) |
| | **Internal (model) Uncertainty** | **ssp126** | | 0.47 (0.91) | 0.47 (0.79) | 0.46 (0.78) | 0.42 (0.80) |
| | | **ssp245** | 0.43 (1.02) | 0.47 (0.96) | 0.49 (0.82) | 0.49 (0.80) | 0.47 (0.79) |
| | | **ssp585** | | 0.50 (0.90) | 0.51 (0.94) | 0.52 (1.00) | 0.53 (1.00) |
| | | **Average** | 0.43 (1.02) | 0.48 (0.93) | 0.49 (0.87) | 0.49 (0.88) | 0.48 (0.88) |


**Table 2-** CMIP6 multi-model mean sink anomalies (with ranges in parentheses) relative to 1995-2015 mean (in mol-C m$^{-2}$ yr$^{-1}$) and internal variability (with model uncertainty in parentheses) for the three scenarios and their average values in NE Pacific and NW Atlantic.

Consistent with the sink anomaly maps (Fig. 1), the regions that show highest uncertainty for any of the sources in the future, are the same regions that show the largest uncertainties in the historical period (Fig. 5). More importantly, the regions of largest future uptake uncertainty are highly correlated with the historical regions of largest uptake (relative to the pre-industrial ocean), as shown by the pattern correlation coefficients above each panel. This is an important finding, because it suggests that knowledge of the regions of modern day surface carbon flux anomaly provides us with information about regions of future uptake uncertainty.

Internal variability from CanESM5 is most dominant in mid-latitude eastern boundary upwelling regions and their extensions, in the North Atlantic, in the western boundary currents of the Gulf Stream and Kuroshio and their extensions, and in the Southern Ocean (Fig. 5). There is wide agreement between different models and estimation methods on regions of largest internal variability (Fig. S4). The regions of large internal variability are correlated with the same highly active regions for the sink anomalies discussed earlier (Fig 1c). This is consistent with McKinley et al. (2017) who argue that modeling and observational studies show that the primary driver of variability in the ocean carbon uptake is ocean circulation and ventilation of the deep ocean. However, correlation coefficients between internal variability and historical uptake are lower than those seen for scenario and model uncertainty. An increase in internal variability with time is seen mostly in the Southern Ocean, the Arctic Ocean, and boundaries of the gyre systems, while the rest of the ocean does not show a clear change. The maps in Figure 5 are averaged over the three scenarios, which masks the changes to some extent. However, we show in the Supplements (see section S2) that changes in the globally averaged internal variability with time are different for different scenarios.

Model uncertainty is consistently highest in the highly active regions (Figure S7), leading to strong correlation with the anomaly maps of Fig. 1c. In these regions, ocean circulation impacts surface $pCO_2$ through advection and water mass transformation regionally (Bopp et al., 2015; Toyama et al., 2017) and models have substantial differences in ocean circulation. Ridge and McKinley (2021) suggest that while global surface carbon fluxes and carbon storage are largely similar across ESMs over the historical period, consistent with the external forcing from atmospheric $pCO_2$ growth being the main driver of the historical sink (McKinley et al., 2020), uncertainties

in ocean circulation may become important in the future under a changing trajectory of atmospheric boundary
conditions. The model uncertainty is largest in the Southern Ocean consistent with CMIP5 models (Frölicher et
al., 2015). Here, mode and intermediate waters are formed, and the complex nature of processes governing the
sinks varies on all time scales (Gruber et al. 2019). Frölicher et al. (2015) note the largest disagreement in ocean
carbon uptake between models is in the Southern Ocean because the exact processes governing heat and carbon
uptake remain poorly understood. The importance of model uncertainty in the Southern Ocean provides a clear
focal point for modelling centers to concentrate their efforts in reducing projection uncertainty.
Scenario uncertainty exhibits the largest change with time. This is by construction as the scenarios deviate from
each other with time to represent a range of pathways for future socio-economic possibilities in order to assess
the long-term impacts of short-term decisions (Riahi et al., 2017). Importantly, the correlation coefficients are
highest between scenario uncertainty and the current regions of large sink anomaly, indicating that the same
highly active regions are the regions that show the largest divergence among scenarios, and that the sink in most
other regions does not respond as strongly to scenario differences. We showed an example of this earlier (Fig. 4),
where the timeseries of the multi-model signals for the three scenarios did not emerge out of internal variability
in the NE Pacific by 2100, whereas they did for the highly active region of the NW Atlantic. This shows that with
$pCO_2$ differences across the air-sea interface being the main driver of the sink (Fay & McKinley, 2013;
Landschützer et al., 2015; Lovenduski et al., 2007; McKinley et al., 2017; 2020), the sink in these active regions
evolves as the atmospheric $CO_2$ concentration changes because ocean processes associated with surface-depth
connectivity constantly dampen the surface ocean $pCO_2$ trend compared with that of the atmosphere. In other
words, the surface water in these regions are constantly renewed, mostly through advection and water mass
formation, with water masses whose $pCO_2$ has not increased at the same rate as the atmosphere. Elsewhere, these
conditions do not hold true and surface water trends match that of the atmosphere, decreasing the sensitivity of
the sink anomaly to the projection scenario. These uncertainties are central to the ability to detect human induced
trends in observations of the surface ocean carbon flux as well as to assess mitigations or make societal decisions,
to which we now turn.

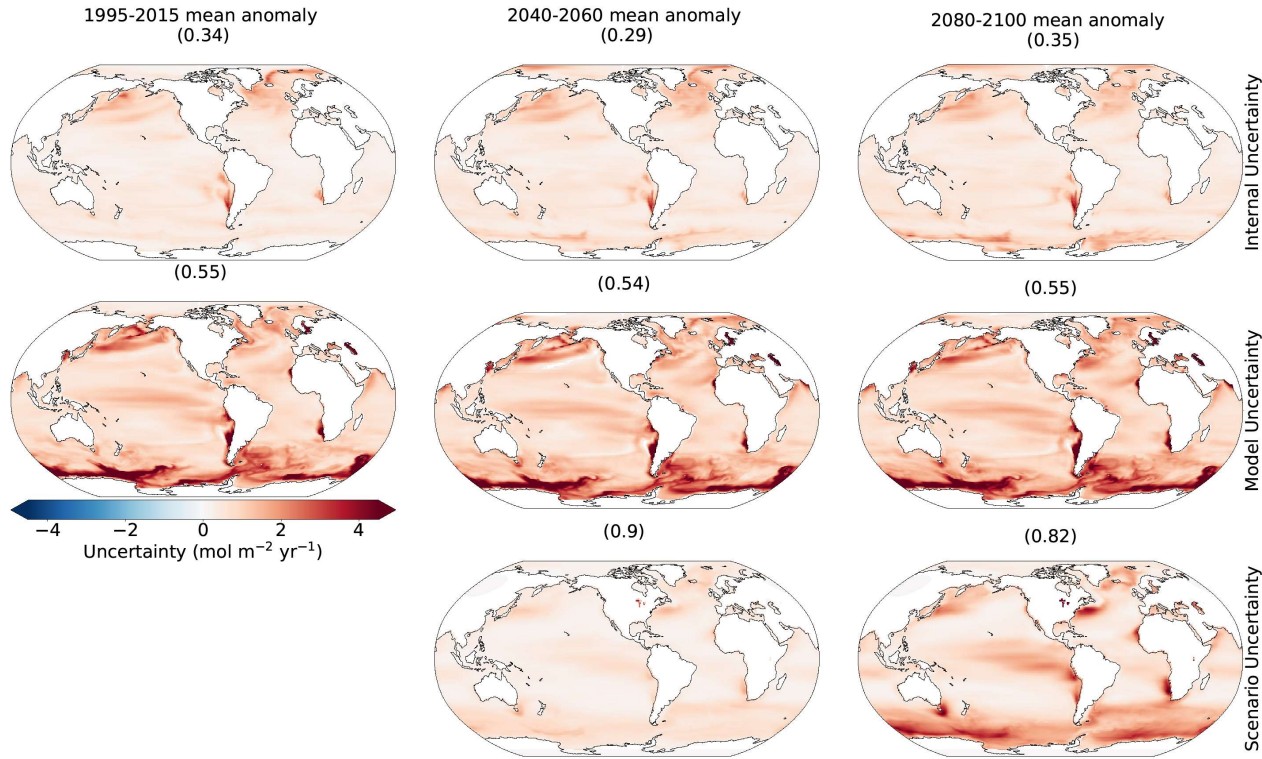


**Figure 5-** Sources of uncertainty averaged over the 20 year mean periods. The rows represent different sources as explained in the methods section at each grid cell. Columns represent different times: the recent (1995-2015), mid-century (2040-2060), and late-century (2080-2100) anomalies relative to the 1995-2015 mean. The numbers are correlation coefficients of each map with the 1995-2015 mean anomaly relative to the 1850-1900 mean (Fig. 1c).

3.4 Detectability

Detectability refers to the ability to robustly identify a forced signal, above and beyond the noise induced by internal climate variability. Previous studies have largely presented a single time of emergence (Lovenduski et al. 2016, Schlunegger et al., 2019, McKinley et al., 2016). However, understanding the regional differences, timescales, and scenario dependence in the detectability of human induced trends in the ocean surface carbon flux is important for informing observational strategies that aim to measure these changes.

We measure the detectability of the CMIP6 multi-model ensemble mean ocean surface carbon flux anomaly using
the time of emergence at each grid point. We use this finest scale as it is the most applicable to observational
communities for sampling. The time of emergence is defined as the point at which the forced signal, given by the
multi-model ensemble mean flux anomaly, relative to 1995-2015, emerges from internal variability, given by the
CanESM5 SMILE.

The signal in human induced surface ocean carbon flux emerges beyond the internal variability earlier in the highly
active regions than anywhere else. This is evident in the Equatorial Pacific, Southern Ocean, the western boundary
currents of the gyre systems, and their extensions (Fig. 6). Ocean regions such as the centres of the mid-latitude
gyre systems and the NE Pacific show late emergence times and, in some cases, no detectability of the signal in
any of the scenarios by 2100. Convergent large-scale circulation and strong stratification in these regions isolates
the surface from the deep ocean limiting their capacity to accelerate their uptake of anthropogenic carbon
(McKinley et al., 2016). An absence of mechanisms constantly drawing surface ocean $CO_2$ trends out of
equilibrium with atmospheric $CO_2$ lets the surface water adjust to the atmospheric trend on short time scales.
Significant changes thus do not take place in the sink as the atmospheric $CO_2$ levels change and scenario uncertainty
is lowest in the same regions (see Fig. 4). This is consistent with the results from Sect. 3.3, in which we showed
that internal variability is a significant source of uncertainty throughout the century in the NE Pacific, with scenarios
never emerging out of the range of internal variability (Fig. 4a, b). Our results for the broad patterns in the multi-
model mean TOE are largely consistent with previous studies, suggesting they are robust and insensitive to the
method of estimating internal variability.  These include studies with single model large ensembles such as
McKinley et al., (2016) that assumed time/scenario independent internal variability, and CMIP5 models such as
Schlunegger et al., (2020) that used only high emission scenario internal variability from four large ensembles to
show there is strong agreement between LEs TOE both locally and spatially. Our results argue that observational
records inside highly active regions are likely sufficient to detect human influence on the ocean carbon sink in the
coming years/decades (2030-2050) if not earlier. Meanwhile, they imply that observational timeseries in quiescent
regions, such as Ocean Station Papa in the NE Pacific, need to interpret any observed trends with care, since internal
variability tends to dominate over human induced trends.



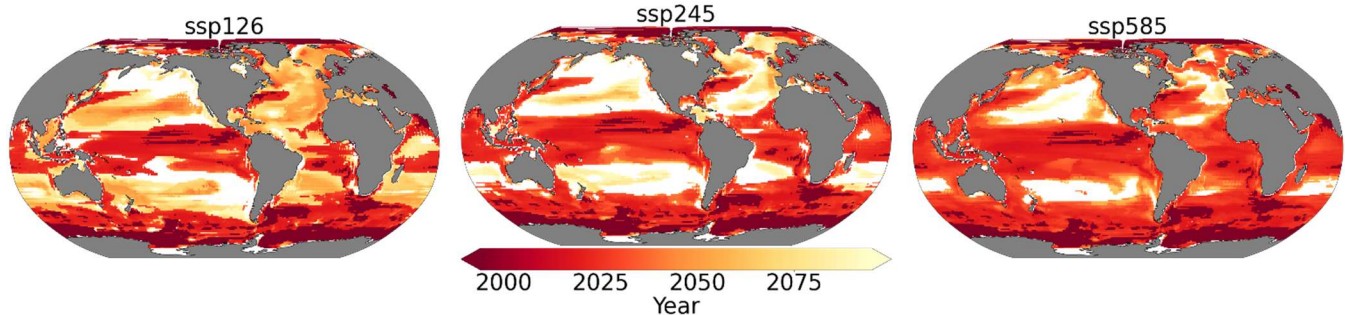


**Figure 6-** Time of emergence of the multi-model mean anomaly under different scenarios. White regions indicate where the anthropogenic signal cannot be detected even towards the end of the century.


Time of emergence strongly depends on the future scenario. Schlunegger et al. (2020) show for two scenarios that modest (~10 yr) TOE differences between different ESMs under strong anthropogenic forcing can evolve into pronounced (60+ yr) TOE differences with moderate mitigation. Here, we make use of three scenarios including a strong-mitigation scenario and account for scenario dependence of internal variability in our approximation using CanESM5. On average, scenarios with smaller forced trends emerge later as the size of the forced trend is critical to the time of emergence (Fig. 2-a). The TOE is earliest on average over the global ocean in ssp585, while it is later in ssp245, and later still in ssp126, consistent with the imposed changes in atmospheric $CO_2$ concentration. The exceptions are quiescent regions that show earlier detectability for ssp126 compared to other scenarios; these exceptions are associated with larger (but negative) anomalies in the latter half of the century under ssp126 which has negative emissions (compare panels d-f, and g-i on Fig. 1). Internal variability does evolve somewhat differently for each scenario, but this is secondary (Fig. S2). Schlunegger et al. (2020) argues that variables such as air-sea $CO_2$ flux which are sufficiently sensitive to emissions emerge early, prior to significant divergence among future scenarios. Consistent with this result, our results indicate that there is broad agreement between scenarios in the TOE patterns, when considering the highly active regions. Interestingly, our scenario-specific TOE shows that differences between scenario TOEs is associated with how sensitive different regions are to emission scenarios. More specifically, comparison to the maps of scenario uncertainty (Fig. 5) shows that TOE differs more across scenarios in regions where scenario uncertainty is small, such as the aforementioned subtropical Ekman convergence regions. Elsewhere, the emergence happens before scenarios diverge significantly. Our results suggest that under the rapidly rising atmospheric $CO_2$ concentrations seen in ssp585, the human signal in the ocean carbon sink will likely be detectable across much of the global ocean over the coming few decades. However, under strong mitigation scenarios, such as ssp126, early emergence (e.g., earlier than 2030) is not expected to occur except in

isolated regions while counter-intuitively, a lower percentage of the global ocean area remains non-emergent by
587 2100.

**4. Conclusions**
Ocean uptake of the increasing atmospheric $CO_2$ in the 21st century is concentrated in a few active regions with 70
percent of the total changes in the sink occurring in less than 40 percent of the global ocean. We analyze the results
from the CMIP6 multi-model mean for the current state of the ocean (1995-2015), and the middle (2040-2060) and
late (2080-2100) 21[st] century relative to the current state for three scenarios. We show that future changes in the
sink are projected to mostly take place within the same historically highly active regions, including the North
Atlantic and Southern Ocean. Our results extend the argument of Wang et al. (2016) that the historical state is a
good predictor of the future state to spatial patterns of change.

We show that the CMIP6 multi-model mean provides a consistent estimate of the spatial patterns of the sink, and
the trend in the sink (globally), compared to the observation-based data product of Landschützer et al. (2016).
These results suggest the CMIP6 models are valid tools for understanding the past and future evolution of the ocean
carbon sink, particularly at broad spatial scales. A notable area of disagreement is that the Landschützer et al. (2016)
data shows larger decadal variability at the global scale than seen in any CMIP6 model or the range of internal
variability from the CanESM5 large ensemble. Gloege et al. (2021) shows that the SOM-FFN method
overestimates the magnitude of decadal variability on the global scale due to the amount of gap filling.

We have shown that the magnitude of uncertainty and its partitioning among different sources differs with scale
and location. On the global scale, scenario uncertainty is the largest source of uncertainty followed by model
uncertainty and internal variability for CMIP6 models. These results are in agreement with previous studies form
the CMIP5 models (Lovenduski et al., 2016; Schlunegger et al., 2020). As the scales of integration (averaging) get
finer, model and internal variability become the dominant sources, respectively. Testing the results on two ocean
regions of about the same size, one in the NE Pacific and one in the NW Atlantic shows that - while consistent with
the results of the scale dependence analysis - the relative importance of the sources of uncertainty also differs with
location. Our test here extends the analysis Schlunegger et al. (2020) with a focus on the association of the location
dependence with whether the regions have highly active carbon sinks. Notably, in highly active regions, such as
the NW Atlantic, scenario uncertainty is large, whereas in more quiescent regions, such as the NE Pacific, internal

variability is more important. The time- and scenario- dependence of internal is another interesting finding that could be the subject of future studies to achieve a better understanding of the driving mechanism and the degree of dependence on future emissions and/or concentrations.

The patterns of high future $CO_2$ uptake uncertainty are highly correlated with the patterns of historical uptake. The correlation coefficients are highest for scenario uncertainty, indicating that the highly active regions have the potential for the sink to evolve according to the atmospheric $CO_2$ concentration, while the rest of the ocean basins do not respond strongly to changes in atmospheric $CO_2$ represented by the different scenarios. This finding has implications for assessment of mitigation and effects of socioeconomic decisions. Our results here are significant in that they show that regions of future uncertainty are strongly associated with known regions of large historical uptake.

Patterns seen in the time-of-emergence have implications for observational campaigns for detection of a signal (Schlunegger et al. 2019; 2020).There is a reverse association between how sensitive a region is to scenario differences (apparent in the scenario uncertainty patterns) and how sensitive the TOE is to scenarios. Our results show that caution should be taken in interpreting the observed changes in regions such as the NE Pacific associated with late emergence of the signal from the decadal (internal) variability. On the other hand, consistent observations in regions such as the Equatorial Pacific, the Gulf Stream and Kuroshio and their extensions, and the Southern Ocean, are likely to detect the emergence of the forced signal out of internal variability earlier in time. Additionally, the patterns in sources of uncertainty show that model uncertainty is largest in the Southern Ocean, consistent with Frölicher et al., 2015. The sink in the Southern Ocean is driven by complex mechanisms involving coupled ocean-atmosphere-ice interactions that require better representation in ocean biogeochemical models. Significant progress in reducing uncertainties can be expected from new methods of bringing together models and observations (Frölicher et al. 2016). Our results provide a motivation to focus modelling as well as observational efforts on the known highly active regions of historical uptake.

Finally, we have shown that internal variability shows clear changes in time and depends on the scenario. The emergence of Large Ensembles (LEs) allows for quantification of these variations if enough ensemble members are available to fully capture internal variability using realizations that start from different initial conditions. Our use of the CanESM5 LE allows for us to account for the nonstationary of internal variability in time, like in Schlunegger et al. (2020), but with the advantage of also accounting for scenario dependence. Model

intercomparison indicates that ESMs show differences in natural variability (Schlunegger et al. 2020). Nonetheless,
our analysis of the global scale, of scale dependence, and of the patterns seen in Time of Emergence are consistent
with previous studies, despite the potential sensitivity to the use of CanESM5 LE. Our methodology to correct for
internal variability from model spread, without filtering or having a large ensemble for each ESM (which would
limit the number of ESMs that can be included and, consequently, underestimate model uncertainty) lays the
foundation for future studies when LEs are available from more ESMs and suggests a need for more modelling
groups to provide such LEs in order to achieve a more robust estimate of internal variability across different ESMs.

## Data Availability

The data used in this study is part of the World Climate Research Programme's (WCRP) 6th Coupled Model
Intercomparison Project (CMIP6) open access data. For details on accessibility see section S1 in the Supplements.
The SOM-FFN data (Landschützer et al., 2017) from Landschützer (2016) can be accessed through the National
Oceanographic Data Center (NODC, https://www.nodc.noaa.gov/archive/arc0105/0160558/3.3/data/0-data/)
operated by the National Oceanic and Atmospheric Administration (NOAA) of the U.S. Department of Commerce.

## Author Contribution

Parsa Gooya conducted the formal analysis, visualization, and original draft preparation. Conceptualization, and
methodology development and validation were a collaboration of the three authors, mainly developed by Parsa
Gooya with contributions from Neil Swart in development, validation, and revision and Roberta Hamme in
validation and revision. Neil Swart and Roberta Hamme provided supervision and reviewing and editing of the
manuscript and methodology. Funding acquisition was carried out by Roberta Hamme.

## Competing of interest

The authors declare that they have no conflict of interest.

**Acknowledgments**

This work was supported by the Marine Carbon Sink project, funded by the Natural Sciences and Engineering Research Council of Canada through the Advancing Climate Change Science in Canada program. We thank Jim Christian for helpful suggestions on a draft of the manuscript.

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
