# Peer review of "Time varying changes and uncertainties in the CMIP6 ocean carbon sink from global to local scale"

_Earth System Dynamics, 2022_

## Author Comment (AC1)

The authors investigate the future ocean carbon sink in CMIP6 models under several SSP scenarios. They quantify the uncertainty in the future sink as a function of model, internal and scenario uncertainty for the global and regional scales. They calculate time of emergence for the forced signal to emerge from internal variability. They find that the future ocean carbon sink is most uncertain in regions of currently highest flux.

The methods used are solid and, with a few minor exceptions, adequately explained. Reasonable assumptions are made. The Supplementary provides useful additional information.

The paper is a contribution to the literature on the CMIP6 models. Most of the calculations done here have been done before for CMIP5 models in several papers, so this is a useful update. It is appropriate for ESD readership. Conclusions are justified by the analysis

Many thanks for the careful and thorough review of the work and your positive reception. We appreciate the constructive comments and suggestions.

Major Comments

On the whole, the paper lacks depth and clarity in the discussion of mechanisms on the ocean carbon sink and how it should evolve in the future. The references to the literature are also somewhat sparse. I encourage the authors to review some more of the literature and to add more mechanistic discussion and connection to previous studies. To do so will make the paper a more useful contribution. Possibilities include Crisp et al 2022 in Reviews of Geophysics, Ridge and McKinley 2021 in Biogeosciences, Hauck et al. 2020 in Frontiers, McKinley et al. 2020 in AGU Advances, Bushinsky et al 2019 in GBC, Schwinger and Tjiputra 2018 in GRL.

Thank you for the recommendations. We have revisited/reviewed the suggested literature and changes have been made to the manuscript in order to embed the study in the previous literature and provide a more mechanistic discussion where possible. We agree that a more mechanistic approach would make this paper a better contribution to the community and changes have been made to the manuscript to serve that purpose. The following is a summary of the updates to both clarify the novelty of the study as well as include a mechanistic overview:

1. Wider citation of previous relevant studies throughout the paper to show how our results support or are different from previous studies and to emphasize places of novelty in our study. The added citations include; Lorenz, (1969); Somerville, (1987); Sarmiento et al., (1998); Lovenduski et al., (2007); Graven et al., (2012); Fay & McKinley, (2013); Bopp et al., (2015); Landschützer et al., (2015); Frölicher et al., (2015) & (2016); Wang et al., (2016); McKinley et al., (2017); Riahi et al., (2017); Toyama et al., (2017); Bushinsky et al., (2019); Schlunegger et al., (2019) & (2020); McKinley et al. (2020); Hauk et al., (2020); Ridge and McKinley, (2021); Terhaar et al., (2021); Crisp et al. (2022);

2. A new paragraph on detectability using time of emergence reviewing previous studies in the introduction.

3. Overview of mechanisms associated with surface-depth connectivity and how they affect the regional patterns of uptake in the historical period and the future.

4. Discussion of the mechanisms responsible for patterns seen in the maps of the sources of uncertainty; especially large model uncertainty in the Southern Ocean and association between regions of large scenario uncertainty and large uptake.

5. Discussion of the mechanisms driving patterns seen in the time of emergence (TOE), and the association of scenario TOE differences with scenario uncertainty (sensitivity to forcing).

6. Two sections were added to the Supplement. Section S5 provides the details of our analysis for diagnosing the "highly active regions" using a new metric. Section S4 details the test done to compare the multi-model trend with that of the observations.

I also recommend changing away from the "hotspots" terminology. For the carbon sink, this term is often used for very small regions, such as western boundary current mode water formation regions. To use this for all of the Southern Ocean, North Atlantic, etc. is also just not a very good choice of words also since these are large, basin scale regions not "spots".

(Also answering the following comments moved from minor comments below:)

**pg 8 line 7-8**: the "hotspots" terminology is too vague, making it hard for the reader to follow.

**Pg 20, line 53:** "mostly in a few hotspot regions" suggests a few small spots when in fact the ocean carbon sink is diffuse and occurring really everywhere (see figure 1c). See Major Comment.

We can see your point here. We have used "highly active regions" instead in the revised version. Moreover, a new section is added to the Supplement (section S3) to clarify how we classify the "highly active regions" - 70 percent of the total sink occurring in less than 40 percent of the global ocean.

Minor

Pg 2, line 46. Bushinsky et al. 2019  should be added to this list

Done!

Pg 3, line 84. ESMs are based in fundamental equation such a Navier Stokes. Yes, there are many details that differ, but there is also a lot of basis in physics! This statement suggests that models are much more of a potpourri than they actually are. Please add some more discussion to more accurately represent ESMs.

This sentence was deleted: "Each model has a specific way of representing the physical world."

Pg 8, line 95. Need to clarify that SOM-FFN is just one realization, not the forced signal. Of course, it is all that we have, so the comparison to the multimodel mean is reasonable. The authors just need to make sure that the text here helps the reader to understand that observations are not the forced component.

We clarified this in the manuscript: "Unlike in ESMs, the observation-based product only represents the one realization of the real world, which includes internal variation, and is therefore not directly equivalent to the forced signal. However, the comparison to the 20 year mean multi-model mean still informs us about the degree of agreement between the two products."

Table 1. Note in caption that internal is from CanESM only

Done!

Both Tables are difficult to read.  Labels in column 2 are too small. The distinction between the scenarios is not clear enough.

Fair enough. This was not the format of the tables originally. We had to be change it to match submission requirements. The tables will be updated to match all your comments.

Figure 2. correct spelling in words in 2b

Will do! Thanks!

pg 11, line 65-66, "test from Santer et al. (2018)" should be defined in methods

A new section has been added to the Supplement.

pg 11, line 69. Strike "in the models" and replace with "in CanESM"

Done!

pg 12, line 77. Figure 2 (and also Figure 4) makes it evident that "model uncertainty" is much the mean spread across the models. Please mention this connection explicitly.

We could not quite understand what your comment refers to. There might be a typo in the comment here.

Pg 17, line 80-83. This section is poorly worded. Please rephrase to avoid "are mostly within hotspots but are not confined to them and do not include all of them".. and to be more specific about the regions to which you refer.

The phrase "The regions of high internal variability (eastern boundary upwelling regions, western boundary currents of the Gulf Stream and Kuroshio, their extensions, and the Southern Ocean) are mostly within hotspots but are not confined to them and do not include all of them. This lack of correspondence explains why the correlation coefficients are not high for internal variability. " was replaced with:

"Internal variability from CanESM5 is most dominant in mid-latitude eastern boundary upwelling regions and their extensions, in the North Atlantic, in the western boundary currents of the Gulf Stream and Kuroshio and their extensions, and in the Southern Ocean (Fig. 5). There is wide agreement between different models and estimation methods in regions of largest internal variability (comparison Fig. S4 added to the supplements). The regions of large internal variability are correlated with the same highly active regions for the sink anomalies discussed earlier (Fig 1c). However, correlation coefficients between internal variability and historical uptake are lower than those seen for scenario and model uncertainty."

"Highly active regions" are also described more specifically in the supplement section S3.

Pg 17, line 87-93. This is just the mean sink, i.e. where low anthropogenic carbon is being brought to the surface. These regions continue to be ventilated from the deep and this is why the sink persists. There is no need to invoke teleconnections.

The final sentence on atmospheric telecommunications has been deleted and the paragraph ends like this:

"The model uncertainty is largest in the Southern Ocean consistent with CMIP5 models (Frölicher et al., 2015). Here, mode and intermediate waters are formed, and the complex nature of the sink varies on all time scales (Gruber et al. 2019). Frölicher et al. (2015) note the largest disagreement in ocean carbon uptake between models is in the Southern Ocean because the exact processes governing heat and carbon uptake remain poorly understood. The importance of model uncertainty in the Southern Ocean provides a clear focal point for modelling centers to concentrate their efforts in reducing projection uncertainty. "

Pg 17, Line 95-96. This description of the scenarios is not sufficiently precise. Scenarios are designed primarily to represent potential futures that socio-economic modeling indicates have potential to be realized. Within this range, there is a selection made of a representative pathways that are not too similar. But this is not the same as to say that they are "designed to deviate". See Riahi et al. 2017 http://dx.doi.org/10.1016/j.gloenvcha.2016.05.009

The sentence is replaced with:

"This is by construction as the scenarios deviate from each other with time to represent a range of pathways for future socio-economic possibilities in order to assess the long-term impacts of short-term decisions (Riahi et al., 2017)."

pg 18 line 01-03. please discuss what are these processes.

The discussion was elaborated on as follows:

"This shows that with $pCO_2$ differences across the air-sea interface interface being the main driver of the sink (Fay & McKinley, 2013; Landschützer et al., 2015; Lovenduski et al., 2007; McKinley et al, 2020; McKinley et al., 2017), the sink in these active regions evolves as the atmospheric $CO_2$ concentration changes because ocean processes associated with surface-depth connectivity constantly keep the surface ocean $pCO_2$ out of equilibrium with the atmosphere. In other words, the surface water in these regions are constantly renewed, mostly through advection and water mass formation, with water masses whose $pCO_2$ has not increased at the same rate as the atmosphere. Elsewhere, these conditions do not hold true and water at the surface equilibrates with the atmosphere on shorter time scales, decreasing the sensitivity to the projection scenario. "

pg 19, line 26 "fixed inactive regions"? please clarify. These regions are not "fixed" or "inactive"

"The fixed inactive regions, such as the centers of the mid-latitude gyre systems and the NE Pacific, show late emergence times and, in some cases, no detectability of the signal in any of the scenarios by 2100." was replaced with:

"Ocean regions such as the centres of the mid-latitude gyre systems and the NE Pacific show late emergence times and, in some cases, no detectability of the signal in any of the scenarios by 2100. Convergent large-scale circulation and strong stratification in these regions isolates the surface from the deep ocean reducing their capacity to hold large amounts of carbon (McKinley et al., 2016). An absence of mechanisms constantly drawing surface ocean $CO_2$ out of equilibrium with atmospheric $CO_2$ lets the surface water equilibrate with and adjust to the atmosphere on short time scales. Significant changes thus do not take place in the sink as the atmospheric $CO_2$ levels change and scenario uncertainty is lowest in the same regions (see Fig. 4). "

pg 20, line 41-51. The size of the forced trend is critical in the time of emergence. The scenarios with smaller forced trend emerge later. Please include this in the discussion

Thanks for the suggestion. Included!

Pg 21, line 72. Strike "basins", replace with "regions"

Done!

Supplementary

Below Eqn S4. "Section S2", instead of "Appendix B"

Edited!

Page numbers are needed in the Supplementary

Added.

Citation: https://doi.org/10.5194/esd-2022-19-RC1

---

## Author Comment (AC3)

The authors investigate the sources of uncertainties in projections of air-sea CO2 fluxes in CMIP6 ESMs and also provide estimates on the time of emergence of the forced signal.

I appreciate the time and effort that the authors put into developing this manuscript. However, I cannot recommend the paper in its present form. Even though I appreciate that the authors tackle an important question, namely the relative role of scenario uncertainty, model uncertainty and internal variability, it remained unclear to me what new insights are gained here in how the future ocean carbon sink evolves. As it stands, it is mostly an update of previous literature and analysis, but this time with CMIP6 models. In addition, I also have some concerns in how internal variability is estimated. See all my comments below.

We appreciate and thank you for your detailed review of the paper and constructive comments. It is of great value to know how we can make this paper a better contribution to the literature by addressing your comments and concerns. Hope you find our responses and edits satisfying.

Major comments:

Embedding results into existing literature

The detailed breakdown of uncertainty in the scenario uncertainty, model uncertainty and internal variability in air-sea CO2 flux projections has been done by many others (e.g., (Lovenduski et al., 2016; Schlunegger et al., 2019, 2020)). Here the authors use CMIP6 models instead of CMIP5 models as used in those previous analyses, but the main results are basically the same as for the CMIP5 models: scenario uncertainty dominates at the global scale, followed by model uncertainty and then internal variability; time of emergence is early in high latitudes and in the tropics. If the authors want to publish this paper, the MS needs to include a thorough discussion on how these new CMIP6 results differ from what we already know from CMIP5. Or how it supports those previous findings. For example, the three last paragraphs in the discussion do not contain any single reference. However as mentioned above, many studies already exist who tackle similar questions. I am fine if the purpose of the paper is to give an update with CMIP6 models, but if so, this needs to be clearly stated upfront.

We have worked to embed the results into the literature by citing relevant previous studies more broadly. The aim of this study was to provide a quantification of the changes in the marine carbon sink and its sources of uncertainty based on CMIP6 models with updates in the methodology used and the analysis done compared to previous CMIP5 studies. We make use of an ensemble of 13 models to better capture model uncertainty in the response to different forcing (scenarios) and we use three scenarios to represent a wide range of future possibilities including a strong mitigation scenario. We have clarified and made sure in the abstract, introduction, and throughout the paper, in what way our analysis differs from

the existing literature. For instance, our method to correct for the internal variability included in the spread across one realization of many models in order to estimate model uncertainty without having to apply a filtering or needing a large ensemble for every model is a novel approach. Moreover, we provide a novel analysis of how the three sources of variability change across the full continuum of scales, and a new metric to show how we can quantify the highly active regions for the sink. Finally, we have edited the manuscript to clearly show how our analysis builds on and supports previous studies and where our methods are different by citing the literature more heavily especially in the conclusions and results sections, and updating the discussions section.

The authors also highlight in the abstract that the ocean carbon sink is concentrated in highly active regions. That has been shown by many studies already (e.g. (Sarmiento et al., 1998)). Again, what is the novelty here?

We have now cited the mentioned paper in the manuscript where we bring up the discussion of highly active regions to show how our results support the previous studies. However, as pointed out in the abstract, we found that more than 70% of the change in the sink relative to the pre-industrial era is concentrated in less than 40% of the global ocean. This quantification of the highly active regions is another novel contribution that we now explain more clearly by defining our classification of highly active regions in the revised manuscript and supplement. We have added a new section to the supplement explaining the analysis done to quantify these highly active regions through a new metric. This metric finds the ocean grid cells over which the largest percentage of the integrated global sink is concentrated in the smallest percentage of the global ocean surface. The resulting highly active regions look as follows:

[Figure]

**Figure S7**- Highly active regions for the sink. The rows represent different scenarios and columns represent different time periods. The land is masked with grey color and grid cells outside of the "highly active regions" are hatched.

Calculating internal variability

The authors use one single ESM (i.e., CanESM5) initial condition large ensemble to estimate the internal variability in the air-sea CO2 flux. Whereas I see the benefit of using a large ensemble to estimate internal variability, as for example internal variability may be sensitive to changes in climate change and therefore changes with time, I suspect that the current results (fractional uncertainty and time of emergence) are heavily biased towards the CanESM5 model and how it represents internal variability. I suspect that different CMIP6 models simulate different magnitude of internal variabilities in air-sea CO2 fluxes. Therefore, the fractional uncertainty as well as the time of emergence might be different when using a different model. Therefore, it may make more sense to use piControl simulations from a variety of CMIP6 models (for example the same as used to estimate model uncertainty; if CMIP6 SMILEs are not available) to estimate internal variability and how uncertainties in the internal variability estimates impact the time of emergence and the uncertainty breakdown.

(The following two questions were moved from your minor comments below to here:)

**L77- Data and methods:** You assume that internal variability is well represented by CanESM5.  But this might not be true.

**L28-45- Data and methods:** As explained above, I am not convinced to use one single model to estimate internal variability given that the models simulate a large spread in the magnitude of internal variability.

Thanks for pointing this out. We show in the supplements (Fig. S2) that, in order to use a large ensemble for the estimation of internal variability, the ensemble size must be large enough to sufficiently capture internal variation, and that internal variability has a scenario dependence that should be accounted for. In the ideal case, if every CMIP model provided sufficiently large SMILES for each scenario, an ensemble mean estimate of the variability could be obtained and would represent a best estimate (but still possibly biased compared to the real world). However, only a handful of CMIP6 models produced multiple ensemble members. We selected the CanESM5 SMILE as it is the only model that has a large enough ensemble over the entire timeline and set of experiments to make estimate internal variability robustly and across scenarios.

The use of a single model to estimate the scale of internal variability leads to some uncertainty in our estimates, as models do not agree perfectly with each other on the variability. Nonetheless, over the historical period, variability between large ensembles from three models that have a large enough number of ensemble members is within 10%, on the global scale (Fig S3). Moreover, the general patterns of the magnitude of internal variability are in good agreement across models and are consistent with known regions of high variability in the observed ocean (comparison Figure S4 is added to the supplements). Hence, the uncertainty imposed by using CanESM is very unlikely to influence our conclusions, although the quantitative details will vary. We have addressed possible biases that may arise from this choice throughout the revised manuscript.

In summary, the importance of internal variability decreases dramatically with time for the global ocean where internal variability is averaged out. Our analysis of fractional uncertainties is consistent with previous studies on both short and long time scales (Lovenduski et al.,2016, Schlunegger et al., 2020) on the global scale. Our goal for regional and local scale analysis is to point out the importance of the specific location in addition to the scale which is in agreement with previous studies, and the association between scenario uncertainty and how active the region is regarding the sink, which is valid and insensitive to the use of CanESM5. We agree that the results include characteristics specific to CanESM5 on regional scales. That could change the results for fractional uncertainties in the NE Pacific and NW Atlantic. Still, our tests on regional and local scales serve as an introduction for the need to understand patterns of the signal relative to internal variation and hence the TOE analysis for which our results are consistent with McKinley et al. (2016) and Schlunegger et al. (2020).

To address your concern and specific suggestion, we repeated the TOE analysis based on estimations of internal variability from the piContrtol runs of a subset of our CMIP6 multi-model ensemble for which the control runs were available. This leaves us with 11 models (excluded are CESM2 and NorESM2-LM). The results are presented below in comparison to the original analysis using CanESM5 LE.

[Figure]

**Fig a-** TOE using internal variability estimation based on multi-model mean time variation of the century-long detrended piControl simulations (Frölicher et al, 2016).

[Figure]

**Fig b-** TOE using internal variability estimation based on CanESM5 LE (Original version).

It is apparent from the figures that the broad patterns of earliest emergence occurring in the highly active regions (Equatorial upwelling regions, Southern Ocean, North Atlantic, western boundary flows and their extensions, etc. ) and late/non emergence in the Ekman convergence regions in mid-latitudes are similar between the two. Moreover, the dependence on scenarios due to the strength of the forcing is also consistent between the two. The piControl simulations show earlier emergence and a smaller percentage of non-emergent grid cells on average, due to the smaller internal variability as a result of the stationarity assumption and independence of the forcing (scenario). All being said, the conclusions in the manuscript about the broad patterns and association with the discussed mechanisms are robust, while the magnitudes are somewhat different between the two analyses. This is most apparent in the Southern Ocean where TOEs are larger in piControl-based analysis which is attributed to the larger internal variability in the Southern Ocean in piControl-based simulations compared to CanESM5 LE (comparison Fig. S4 added to supplements).

In conclusion, both the CanESM5-based numbers and the piControl-based numbers are imperfect. However, the clear time/scenario dependence of internal variability and low error range between the available large ensembes makes our CanESM5-based analysis a more reliable estimate of the TOE. In response to your comment, we added this analysis comparing piControl-based TOE with CanESM5-based TOE, as well as a comparison map of different estimates of internal variability over the historical period to a new section in the supplements (section S3). Finally, we use the discussions of 1. the need to account for the change in internal variability with time and scenario, 2. the need for at least a certain number of realizations to capture internal variability in a large ensemble, and 3. the proposed methodology to correct for the internal variability included in model spread without needing to use any filtering (if we have a fair estimate of internal variability), to advocate for the need for more models to provide large ensembles so that the internal variability can be better estimated as the mean spread across multiple Large Ensembles.

Accounting for drift in air-sea CO2 fluxes in CMIP6 ESMs

Did you account for potential drifts in air-sea CO2 fluxes in the piControl simulations of the CMIP6 models? To estimate model uncertainty one should use the difference between the historical-scenario simulation and the piControl simulation (long-term trend). This would also allow to include the CNRM-ESM2-1 model as this model has a relatively large preindustrial outgassing of about -0.75 Pg C/yr, which is the reason why the 'present-day' CO2 flux is below all other models as shown in the Supplementary Figure S1. When correcting for this offset the model is close to all other models.

There are two different arguments here. One is the steady state flux and the other one is the drift. Since we use anomalies in our analysis, the steady state flux is removed. Hence, the correction does not explain why CNRM-ESM2-1 is an outlier. For a possible correction for the large outgassing (sum of different parameters such as model bias, outgassing due to riverine input, etc.), we need to make sure we do not remove a component of the sink that is crucial for comparison to the observation-based product. Given that, a deeper investigation of the model in the control simulations based on the CNRM-ESM2-1 model behavior assessment publications would possibly allow for the correction. However, the evaluation of the performance of individual models is beyond the scope of this paper.

We did not account for the drift and made sure that this is clearly stated in the revised manuscript. However, the drift is known to be small in the models compared to the historical trends (Hauck et al, 2020). Moreover, piControl simulations are not currently available for some of our selected models due to ESGF maintenance issues at some institutions. For 11 of our CMIP6 models for which piControl runs are available, on average, the drift is an order of magnitude smaller than the change in the model scenario with the smallest trend over the 21st century, on the global scale.

Accuracy of text

There are many places in the manuscript where the text is not accurate, and the reader might have difficulties understanding the details. For example, on page 11 l66, you state that 'the trend in ocean carbon sink anomalies are statistically consistent between models and obs-based products based on tests from Santer et al. 2018. What test is this? The method has not been introduced in the method section.

We have included a new section in the supplement to explain the test. Thanks for pointing this out.

Minor comments (FYI: the line numbers are confusing in the MS)

Introduction:

L36-47: This paragraph mixes the description of the anthropogenic flux pattern with the total CO2 flux pattern. This is rather confusing.

The updated paragraph looks like this:

"The ocean's capacity to absorb anthropogenic $CO_2$ is not uniformly distributed (McKinley et al., 2016, Sarmiento et al., 1998). Despite increasing atmospheric $CO_2$ concentrations, the air-sea $CO_2$ flux does not change much in the subtropical gyres. The regions where ocean carbon uptake notably increases are those with strong exchange between the surface and the deep ocean (Ridge and McKinley, 2021; Frölicher et al., 2015; McKinley et al., 2016). This response of the ocean carbon sink to increasing atmospheric $CO_2$ levels consists of changes in both the anthropogenic and the natural carbon sink (Crisp et al. 2022, McKinley et al. 2020, Hauk et al., 2020, Gruber et al. 2019, Frolicher et al, 2015). Even within regions, there are large variations in the dominant mechanisms and the direction of the carbon sink. In the Southern Ocean for instance, the spatial superposition of natural and anthropogenic $CO_2$ fluxes leads to a relatively strong uptake band between approximately 55°S and 35°S (Gruber et al., 2019). However, south of the Polar Front (55°S), the different estimates agree less well (Gruber et al., 2019). Supported by measurements based on biogeochemical floats (Bushinsky et al., 2019; Gray et al., 2018; Williams et al., 2018), Gruber et al. (2019) argue that the region was most likely a small source in 2019. "

L36: I guess there are many older papers that could be cited here (e.g. Takahashi or Sarmiento)

Thank you for the suggestions. We have added the Sarmiento (1998) paper to the citations, however, Takahashi does not talk about the anthropogenic sink specifically as far as our knowledge of the literature goes. Is there any specific paper you have in mind that we can refer to?

L38: Maybe cite here (Frölicher et al., 2015)

Agreed! Added to the citations.

I.52: Laufkötter et al. (2015) does not fit here as they do not look at air-sea $CO_2$ fluxes. Maybe include (Terhaar et al., 2021) instead.

Thanks for the suggestion.

L54-59: Schlunegger et al. (2020) also investigated the sources of uncertainties in air-sea $CO_2$ fluxes. This study should be mentioned here as well.

Edited as follows:

"Projection uncertainty varies with lead time, spatial averaging scale, and from region to region (Lovenduski et al., 2016; Schlunegger et al., 2020)."

L69: 'along with others': can you elaborate a bit what other processes you have in mind here? What about small-scale processes such as eddies, etc.?

The sentence was updated to:

"Modes such as the El Niño–Southern Oscillation, North Atlantic Oscillation, Atlantic Multidecadal Oscillation, Pacific Decadal Oscillation, and Southern Annular Mode (SAM) contribute to this internal variability. Internal variability also includes variability that acts on shorter time and spatial scales, such as submesoscale and mesoscale ocean features (Frolicher et al., 2016)."

I 71: 'beyond short timescales': Please backup this claim with a reference

The Following citations are added:

Somerville, R.C.J. The predictability of weather and climate. *Climatic Change* 11, 239–246 (1987). https://doi.org/10.1007/BF00138802

Lorenz E. N. The predictability of a flow which possesses many scales of motion. *Tellus.* 1969;21:19

I84: not only the physical world but also the biogeochemistry for example.

This sentence was also commented by our other referee and for the sake of accuracy, it was omitted from the manuscript.

L98-99: Can you update these number with CMIP6 estimates?

Wil be updated.

L99-00 (page 4):  The introduction lacks a paragraph on earlier results. For example, Lovenduski et al. (2016), McKinley et al. (2016), and Schlunegger et al. (2020) have already tackled similar questions using CMIP5-type models and SMILEs. This needs to be stated upfront here.

Lovenduski et al. (2016), and McKinley et al. (2016) were cited in the introduction to review the literature. We have added a new paragraph advocating for the need to understand the patterns in TOE and have pointed to these earlier studies again, adding Schlunegger et al. (2020) that was not previously mentioned. Moreover, we have made sure to point out what approach was used in these studies and in what way our methods differ. The paragraph looks as follows (page 4, after the discussion about scenario uncertainty before the road map):

"Together with the patterns of changes in the sink, the patterns of internal variability allow for an assessment of the required timescales for detection of changes in the ocean carbon sink. Detection means that we can robustly separate the forced signal from internal variability (McKinley et al., 2016). Detectability can be assessed using Time of Emergence (TOE; Hawkins and Sutton, 2012; Lovenduski et al., 2016; McKinley et al., 2016; Rodgers et al., 2015; Schlunegger et al., 2020 & 2019). For example, McKinley et al. (2016) and Schlunegger et al. (2019) showed that the forced signal of increasing ocean carbon uptake is not detectable in the Ekman convergence regions of the subtropical gyres. Schlunegger et al. (2020) builds on that using four large ensembles of CMIP5 ESM simulations with two forcing scenarios to show that air-sea $CO_2$ flux TOEs show strong agreement between the large-ensembles not just for global and regional scales but also locally and spatially. Their use of only four models and two scenarios, however, potentially underestimates the contribution of model and scenario uncertainty."

Data and Methods:

L07-15: Did you use CO2 concentration driven simulations or CO2 emission driven simulations. I guess the former, but please clarify.

Edited! We are using concentration driven simulations and this is clearly mentioned in the revised manuscript.

**L54: I am a bit confused here. Do you correct here each model with the internal variability from the CanESM5? But what if the different models have a different internal variability than the CanESM5?**

note: Variance$^*$ refers to the Mathematical function with the same name.

Equation S3 shows that if we decompose each models' one realization signal to the sum of the forced signal and a deviation due to internal variability, then by taking the Variance[*] across the multimodal first realizations (assuming that Covariance between "deviations due to internal variability" and "Forced signal" equals zero), the Variance[*] of the forced signal across many models equals the Variance[*] of the first realizations across the multimodal ensemble minus the Variance[*] of the "deviations from the forced signal due to internal variability" across the same models. Since internal variability is assumed to be random noise, in a large sample, the Variance[*] of the "deviations from the forced signal due to internal variability" across many models equals the internal variability. So, we can correct for internal variability included in the spread across different models' realizations if we have an estimate of internal variability.

The use of the CanESM5 LE as an estimate of internal variability is an assumption here clearly stated in the manuscript. As discussed in the manuscript and in detail earlier in this document, internal variability from CanESM5 LE is a fair approximation for internal variability based on current model data availability to account for nonstationarity in time based on scenario. Finally, we have revised the manuscript to make this correction more comprehensible. Thanks for the comment.

Results

L20: Maybe state in the first sentence of the Figure caption what quantity Figure 1 shows.

Edited!

Figure 2b: uncertainty is wrongly written in the Figure – Typo.

Edited!

Figure 2b: y axis: Fraction of what? Please clarify.

Explained in the caption and the figure is updated.

L66-68: How did you test this? How did you conclude that SOM-FFN shows a larger multidecadal variability? This is unclear?

We included a section on how we compare the trends from SOM-FFN to the multi-model mean in the supplements. The analysis shows the trends are consistent between the two products. Given that, the SOM-FFN smoothed time series (using a 10-year running mean) goes outside of the range of internal variability based on CanESM5 LE, which is on top of

the multi-model mean signal (forced trend). It can also be shown that the variance of the SOM-FFN time-series is higher than any individual realization from the models.

We have included in both the text and Fig. 2 caption that all time-series are plotted after applying a 10-year running mean filter.

L77-78: Isn't that obvious, given that scenario uncertainty is zero over the historical period?

Yes, it is. The point here however was that in the historical period model uncertainty is a larger source of uncertainty compared to internal variability, and in the future scenario uncertainty starts to grow larger and larger until it becomes the dominant source.

L92-93: Schlunegger et al. (2020) shows it for many more regions. See their Figure 9.

Schlunegger et al. (2020) talk about how the partitioning of uncertainty differs from region to region and does not mention the magnitude of uncertainty. Still, we have referenced this study in the paragraph where it was relevant. Thanks for the suggestion.

L77-78: 'highly significant finding'. This has been shown already in (Wang et al., 2016). They show that models that simulate a small ocean anthropogenic carbon uptake over the last decades also simulate a small uptake over the 21st century.

Wang et al. (2016) discuss how the projected air-sea $CO_2$ fluxes are strongly associated with the simulated air-sea $CO_2$ fluxes in the historical condition, confirming that the historical state is a good predictor for the future state. We have cited their results in the revised manuscript to better back up our findings. However, throughout the paper, when we are talking about correlations, we are measuring spatial pattern correlations in contrast to Wang et al. (2016) estimation of magnitude correlations. Moreover, the phrase "highly significant finding" that you pointed out is referring to the pattern correlations between projected model/scenario uncertainty and the modern day flux anomalies, not the flux anomalies themselves. This is to mention that not only from Wang et al. (2016) we know that the historical state is a good predictor for the future state, we know it also informs us about the regions (patterns) of future uptake and model/scenario uncertainty.

L89-91: The large uncertainty in simulated uptake of Cant in the Southern Ocean simulated by ESMs has already been highlighted in previous studies (e.g. Frölicher et al. 2015)

We have made sure to cite the relevant literature with which the results from CMIP6 in this study are consistent. The edited section looks like this:

"The model uncertainty is largest in the Southern Ocean consistent with CMIP5 models (Frölicher et al., 2015). Here, mode and intermediate waters are formed, and the complex nature of the sink varies on all time scales (Gruber et al. 2019). Frölicher et al. (2015) note the largest disagreement in ocean carbon uptake between models is in the Southern Ocean

because the exact processes governing heat and carbon uptake remain poorly understood. The importance of model uncertainty in the Southern Ocean provides a clear focal point for modelling centers to concentrate their efforts in reducing projection uncertainty. "

L41: Which previous studies? Please clarify.

The statement "Previous studies have largely presented a single time of emergence; however, the time of emergence strongly depends on the future scenario." is updated to:

"Time of emergence strongly depends on the future scenario. Schlunegger et al. (2020) show for two scenarios that modest (~10 yr) TOE differences between different ESMs under strong anthropogenic forcing can evolve into pronounced (60+ yr) TOE differences with moderate mitigation. Here, we make use of three scenarios including a strong-mitigation scenario and account for scenario dependence of internal variability in our approximation using CanESM5. On average, scenarios with smaller forced trends emerge later as the size of the forced trend is critical to the time of emergence (Fig. 2-a)."

Discussions:

L63-64: Where is this shown.?There is no formal analysis on that in the paper.

Explanation added to the methods.

L69-97: All three paragraphs lack of any reference, even though many studies have investigated similar questions in the past. This needs to be changed.

The whole section is updated in this way:

[revised manuscript text omitted]

---

## Author Response (AR2)

Dear colleagues,

your revised manuscript went through review round yielding very conflicting responses. I therefore searched for a third, independent assessment, which took extra time. Based on this additional feedback, I am happy to inform you that your manuscript can now be published subject to minor revisions with editor review only.

We thank you and appreciate your detailed review of the manuscript which helps us make this a better contribution to the community. Here, we explain how we have addressed the reviewers' comments.

Please address the specific comments of reviewer #2:
* * *
"The authors have added references and more mechanistic description to the paper, as requested. This emphasizes the lack of novelty in the work. I have also taken the time to review the original review of Reviewer 2, and I concur with their finding that this paper is not providing new insights. The paper repeats analyses already presented, with the only distinction being that CMIP5 or a large ensembles was analyzed previously while here CMIP6 is analyzed. The update of model versions would not be expected to substantially change the behavior of the ocean carbon sink, and the authors show that this expectation is fulfilled. The specific calculation of "70% change in 40% of the area" is new, but is not particularly interesting because it has been know that some regions are intense carbon sinks has long been known.

We thank the reviewer for their time reviewing the revised manuscript. Our work does replicate and build on previous studies, but it also contains novel aspects. We update previous results with the new CMIP6 dataset, confirming that previous CMIP5 studies still hold. We also introduce new methodologies, including our large ensemble approach, and novel analyses, including our representation of sources of uncertainty across spatial scales. Our work brings together different ideas and techniques, previously tested separately, and frames them into a single study that addresses the ocean carbon sink based on the most recent version of CMIP models using a wide range of models and scenarios. Moreover, it provides quantification on different ideas that were known collectively through a combination of previous studies (such as our diagnosing the highly active regions, and performing a scale dependence analysis over a continuum of scales)  into a coherent storyline that connects different pieces to each other.

Minor Comments:

Line 39-40: over what timeframe does this sentence apply?
Updated to:

"Despite increasing atmospheric $CO_2$ concentrations, the air-sea $CO_2$ flux does not change much in the middle of the subtropical gyres over the century starting in 1990. "

Line 42-44: This is incorrect. By definition, the response to increasing atm CO2 is, by definition, the anthropogenic sink.
Thanks for pointing this out. We would like to bring your attention to the following lines from Hauck et al. (2020):

"Note that this definition of the ocean carbon sink SOCEAN in the GCB is different from the definition of the "anthropogenic CO2 sink" referred to as the change in ocean carbon content only due to the direct effect of increasing CO2 concentration in the atmosphere ($F_{ant,ss}+F_{ant,ns}$), often used in the observational ocean carbon cycle community (e.g., Gruber et al., 2019)."

We have clarified the wording here in the manuscript to emphasize the direct absorption response and the changes to the natural background fluxes:

"The response of the ocean carbon sink to increasing atmospheric $CO_2$ levels consists of a direct absorption response as well as climate change induced perturbations to the natural background carbon fluxes (Crisp et al. 2022, McKinley et al. 2020, Hauk et al., 2020, Gruber et al. 2019, Frolicher at al, 2015). "

Line 47: which "different estimates"?
Added "(Landschützer et al., 2016,Gruber et al., 2009, Takahashi et al., 2009)"

Line 254: This "metric" is percent change in sink per area? Please be explicit as to what this metric is. Please also explain why it is useful, and explain of what this is a "metric". Most importantly, what supposed to be learned by comparing models using this "metric" or by tracking it over time?

The metric is a criterion to identify the highly active regions. We have added a brief description of the criterion to the main text and refer the reader to the supplement for further information:

"Here, we provide a new criterion for identifying these highly active regions based on comparing the integrated global sink anomaly within grid cells above a certain threshold to the percentage of ocean area they occupy (see Supplement S5)."

In summary, this metric is a threshold that can *diagnose* the highly active grid cells from the rest of ocean grid cells, without having to make assumptions about the boundaries of the active regions for the sink. The resulting regions from this analysis match the regions of intense uptake change (trends) described in previous studies. To clarify its usefulness, we have added to the supplement: "The evolution of the threshold value with time, and the corresponding boundaries of the highly active regions, have implications for the evolution and efficiency of ocean carbon sink under the changing climate."

Line 398-399: The word "active" is used twice. Please don't use the word "active" to explain "active".
In the mentioned lines, the word active is used twice to emphasize the reason for the term "highly active regions" - they actively respond to increasing atmospheric $CO_2$.

Figure 4. Please provide a figure title that immediately identifies this as NE Pacific and NW Atlantic . Please also mark NE Pacific and NW Atlantic on figures themselves.

Figure 4 shows essentially the same information as Figure 5. I don't think Figure 4 is needed.

Thanks for pointing this out. We have updated the figure to label the regions in NE Pacific and NW Atlantic for each row of panels. We prefer to keep both Figure 4 and 5, because:

1- Figure 4 shows averages over specific regions in figure 5 which are not easily comparable visually on Figure 5;
2- Figure 5 alone does not reflect the effects of averaging on the signals as well as on the uncertainty;

3- Figure 4 provides intuition on comparing highly active and and not active regions as well as the association of scenario uncertainty with this designation using time-series and uncertainty ranges.
4- The removal of the figure would compromise the storyline.

Line 545. McKinley et al did not study CMIP5. They studied CESM-LE. Please correct Throughout

Thanks for pointing that out. They use both an ensemble of CMIP5 models and CESM-LE under the RCPs defined through CMIP5. We have clarified this throughout the paper.
* * *
And also the comments of reviewer #3:
* * *
The manuscript presents an analysis of uncertainties in ocean carbon uptake using methods that have previously been applied elsewhere to similar questions. The authors have nevertheless presented sufficiently interesting results that this work should be publishable as an incremental advance. However, I do believe that there are several points where there are misinterpretations of the model results, and misunderstanding of the connection to broader community efforts, and I believe that these critical (but minor by standards of revisions) issues should be clarified and regularized before the manuscript can be published.

We are thankful for the reviewer's constructive comments and welcoming review. We have addressed their specific concerns and comments as follows.

Overall there are two general arcs to the study, one geared towards quantitative analysis of uncertainty, and one geared towards interpretations and recommendations based on the analysis. It is the latter (the implications and interpretations) that are in particular need of attention. On Page 8 (lines 4-8) the authors state that the "largest change takes place in regions such as the North Atlantic, ….These regions of largest changes in the carbon sink seem to be the regions where there is a surface-depth connectivity. We refer to these regions as hotspots".

Whereas I fully agree with the authors that there are subduction hotspots where there is unambiguously a surface-interior exchange of properties via subduction, this in no way provides justification for arguing that gas exchange is dominated by these regions. As has been demonstrated by Iuidcione et al. (2016; Sci. Rep.) using GLODAP data, approximately 60% of anthropogenic carbon enters the ocean over the broad expanses of the subtropical cells, rather than in the hotspot regions highlighted here. This was also shown in the study of Rodgers et al. (JClim, 2020) with a CMIP5-class ocean model, where 60% of

anthropogenic carbon uptake occurs over 45S-45N. Certainly there is enhanced uptake over western boundary current regions etc., but when uptake is analyzed in terms of density (Iudicone et al., 2016), models indicate that this enhancement of CO2 fluxes in western boundary currents is not equivalent to dominance in uptake.

As a related point, the authors state in the second paragraph of the Introduction (page 2, lines 36-38) that the air-sea flux doesn't change much over the subtropical gyres, with strong uptake being confined to regions with strong subduction. Again, in models where this has been evaluated (Iudicone et al., 2016) this is not true, so if the authors wish to make this point the burden is on them to demonstrate where previous studies are wrong.

Thanks for pointing out these interesting studies. We removed the term "hotspot" in our previous revision to avoid confusion, changing this to "highly active region".

More importantly, our discussion is not inconsistent with these previous studies. Ludicone et al. (2016) and Rodgers et al. (2020) describe values over the entire region of tropics and subtropics (including the western boundary currents), while we discuss more specific regional patterns. Importantly, there are large regions within the tropics and subtropics that are included in what we refer to as highly active regions (see Fig. S7). That said, Fig. 4 from Rodgers et al. (2020) is also showing that regionally, when compared to the pre-industrial state, the biogeochemically coupled model do not detect large changes except that they are concentrated at some regions which consistently match the regions we are *diagnosing* and describing as highly active (also shown in McKinley et al; 2016, Froelicher et al; 2015, etc.). Of course, a variety of combinations of ocean grid cells (regions) can explain 60% of the total anthropogenic uptake. What we focus on here is to account for the smaller scale regional differences when quantifying the highly active regions by also considering the area covered. Additionally, our diagnosis of highly active regions refers to their evolution in uptake as atmospheric $CO_2$ concentration increases, not to the sink itself. In other words, we focus on where the sink is accelerating rather than its absolute magnitude.

Finally, we have reviewed the manuscript and made sure that the text is not ambiguous in distinguishing uptake anomalies from the uptake itself, and makes clear where patterns of regional changes are considered and when averaging is taking place. Additionally, the lines in the introduction that the reviewer specifically pointed to were clarified as follows:

"Despite increasing atmospheric $CO_2$ concentrations, projected air-sea $CO_2$ fluxes do not change much in the middle of the subtropical gyres over the decade starting in 1990. The regions where ocean carbon uptake notably increases are those with strong exchange between the surface and the deep ocean (Ridge and McKinley, 2021; Frölicher et al., 2015; McKinley et al., 2016). "

Most importantly, the authors use some of these untested inferences to then propose that future optimization of the ocean observing system should be focused on the regions with large natural variability in carbon uptake. This is problematic for a number of reasons. First

and foremost, it is not based on a quantitive assessment of observing system design for carbon. To do this, I believe the authors would be obliged to conduct an observing system simulation experiment (OSSE) as part of their study Second, even if an OSSE were to be conducted and demonstrate this for the case of carbon (which I do not believe would be the outcome of an OSSE), the authors really should refer back to the justification given through GOOS and others involved in planning for an optimized ocean observing system for ocean biogeochemistry, and make sure to reference correctly how any recommendations here fit their broader goal, if the authors intend to make such recommendations.

My recommendation is that the authors consider scaling back on their recommendations, by at least carefully qualifying them, in terms of community priorities and the fact that an OSSE has not been considered here. I think that the rest of the scientific presentation could stand in that case, and satisfy the requirements for sufficient incremental advance to warrant publication.

We can see the reviewer's point here and appreciate their concern. We have addressed their specific suggestions and scaled back on our recommendations, by pointing out only what our study explicitly suggests, acknowledging that we have not conducted an OSSE and pointing out caveats instead of explicit recommendations. We went through the manuscript and made changes accordingly. These include but are not limited to:

In the abstract: the finishing lines were changed to:

"In agreement with CMIP5 studies, our results suggest that to for a better chance of early detection of changes in the ocean carbon sink, and to efficiently reduce uncertainty in future carbon uptake, highly active regions, including the Northwest Atlantic and the Southern Ocean, should receive additional focus for modeling and observational efforts."

In section 3.4: Second paragraph, lines 17-19 updated to:

"Our results argue that focusing observational records inside efforts on the highly active regions are likely sufficient in order to detect human influence on the ocean carbon sink in the coming years/decades (2030-2050) if not earlier."

In conclusions:

-   In the first paragraph, the following line was deleted:

This result implies that known regions of high historical uptake, are the same regions to prioritize for observing the future evolution of the sink.

-   In the fifth paragraph, lines 6-8 were updated to:

"On the other hand, consistent observations in regions such as the Equatorial Pacific, the Gulf Stream and Kuroshio and their extensions, and the Southern Ocean, are likely to detect the emergence of the forced signal out of internal variability earlier in time."